# LARGE BATCH OPTIMIZATION FOR DEEP LEARNING: TRAINING BERT IN 76 MINUTES

**Yang You**[2], **Jing Li**[1], **Sashank Reddi**[1], **Jonathan Hseu**[1], **Sanjiv Kumar**[1], **Srinadh Bhojanapalli**[1]
**Xiaodan Song**[1], **James Demmel**[2], **Kurt Keutzer**[2], **Cho-Jui Hsieh**[1,3]

Yang You was a student researcher at Google Brain. This project was done when he was at Google Brain.

Google[1], UC Berkeley[2], UCLA[3]

{youyang, demmel, keutzer}@cs.berkeley.edu, {jingli, sashank, jhseu, sanjivk, bsrinadh, xiaodansong, chojui}@google.com

## ABSTRACT

Training large deep neural networks on massive datasets is computationally very challenging. There has been recent surge in interest in using *large batch* stochastic optimization methods to tackle this issue. The most prominent algorithm in this line of research is LARS, which by employing *layerwise adaptive* learning rates trains RESNET on ImageNet in a few minutes. However, LARS performs poorly for attention models like BERT, indicating that its performance gains are *not* consistent across tasks. In this paper, we first study a principled layerwise adaptation strategy to accelerate training of deep neural networks using large mini-batches. Using this strategy, we develop a new layerwise adaptive large batch optimization technique called LAMB; we then provide convergence analysis of LAMB as well as LARS, showing convergence to a stationary point in general nonconvex settings. Our empirical results demonstrate the superior performance of LAMB across various tasks such as BERT and RESNET-50 training with very little hyperparameter tuning. In particular, for BERT training, our optimizer enables use of very large batch sizes of 32868 without any degradation of performance. By increasing the batch size to the memory limit of a TPUv3 Pod, BERT training time can be reduced from 3 days to just 76 minutes (Table 1). The LAMB implementation is available online[1].

## 1 INTRODUCTION

With the advent of large scale datasets, training large deep neural networks, even using computationally efficient optimization methods like Stochastic gradient descent (SGD), has become particularly challenging. For instance, training state-of-the-art deep learning models like BERT and ResNet-50 takes 3 days on 16 TPUv3 chips and 29 hours on 8 Tesla P100 gpus respectively (Devlin et al., 2018; He et al., 2016). Thus, there is a growing interest to develop optimization solutions to tackle this critical issue. The goal of this paper is to investigate and develop optimization techniques to accelerate training large deep neural networks, mostly focusing on approaches based on variants of SGD.

Methods based on SGD iteratively update the parameters of the model by moving them in a scaled (negative) direction of the gradient calculated on a minibatch. However, SGD's scalability is limited by its inherent sequential nature. Owing to this limitation, traditional approaches to improve SGD training time in the context of deep learning largely resort to distributed asynchronous setup (Dean et al., 2012; Recht et al., 2011). However, the implicit staleness introduced due to the asynchrony limits the parallelization of the approach, often leading to degraded performance. The feasibility of computing gradient on *large minibatches* in parallel due to recent hardware advances has seen the resurgence of simply using synchronous SGD with large minibatches as an alternative to asynchronous SGD. However, naïvely increasing the batch size typically results in degradation of generalization performance and reduces computational benefits (Goyal et al., 2017).

Synchronous SGD on large minibatches benefits from reduced variance of the stochastic gradients used in SGD. This allows one to use much larger learning rates in SGD, typically of the order square root of the minibatch size. Surprisingly, recent works have demonstrated that up to certain minibatch sizes, linear scaling of the learning rate with minibatch size can be used to further speed up the

---

[1]`https://github.com/tensorflow/addons/blob/master/tensorflow_addons/optimizers/lamb.py`

training Goyal et al. (2017). These works also elucidate two interesting aspects to enable the use of linear scaling in large batch synchronous SGD: (i) linear scaling of learning rate is harmful during the initial phase; thus, a hand-tuned warmup strategy of slowly increasing the learning rate needs to be used initially, and (ii) linear scaling of learning rate can be detrimental beyond a certain batch size. Using these tricks, Goyal et al. (2017) was able to drastically reduce the training time of ResNet-50 model from 29 hours to 1 hour using a batch size of 8192. While these works demonstrate the feasibility of this strategy for reducing the wall time for training large deep neural networks, they also highlight the need for an adaptive learning rate mechanism for large batch learning.

Variants of SGD using layerwise adaptive learning rates have been recently proposed to address this problem. The most successful in this line of research is the LARS algorithm (You et al., 2017), which was initially proposed for training RESNET. Using LARS, ResNet-50 can be trained on ImageNet in just a few minutes! However, it has been observed that its performance gains are *not* consistent across tasks. For instance, LARS performs poorly for attention models like BERT. Furthermore, theoretical understanding of the adaptation employed in LARS is largely missing. To this end, we study and develop new approaches specially catered to the large batch setting of our interest.

**Contributions.** More specifically, we make the following main contributions in this paper.

- Inspired by LARS, we investigate a general adaptation strategy specially catered to large batch learning and provide intuition for the strategy.

- Based on the adaptation strategy, we develop a new optimization algorithm (LAMB) for achieving adaptivity of learning rate in SGD. Furthermore, we provide convergence analysis for both LARS and LAMB to achieve a stationary point in nonconvex settings. We highlight the benefits of using these methods for large batch settings.

- We demonstrate the strong empirical performance of LAMB across several challenging tasks. Using LAMB we scale the batch size in training BERT to more than 32k without degrading the performance; thereby, cutting the time down from 3 days to 76 minutes. Ours is the first work to reduce BERT training wall time to less than couple of hours.

- We also demonstrate the efficiency of LAMB for training state-of-the-art image classification models like RESNET. To the best of our knowledge, ours is first adaptive solver that can achieve state-of-the-art accuracy for RESNET-50 as adaptive solvers like Adam fail to obtain the accuracy of SGD with momentum for these tasks.

## 1.1 RELATED WORK

The literature on optimization for machine learning is vast and hence, we restrict our attention to the most relevant works here. Earlier works on large batch optimization for machine learning mostly focused on convex models, benefiting by a factor of square root of batch size using appropriately large learning rate. Similar results can be shown for nonconvex settings wherein using larger minibatches improves the convergence to stationary points; albeit at the cost of extra computation. However, several important concerns were raised with respect to generalization and computational performance in large batch nonconvex settings. It was observed that training with extremely large batch was difficult (Keskar et al., 2016; Hoffer et al., 2017). Thus, several prior works carefully hand-tune training hyper-parameters, like learning rate and momentum, to avoid degradation of generalization performance (Goyal et al., 2017; Li, 2017; You et al., 2018; Shallue et al., 2018).

(Krizhevsky, 2014) empirically found that simply scaling the learning rate linearly with respect to batch size works better up to certain batch sizes. To avoid optimization instability due to linear scaling of learning rate, Goyal et al. (2017) proposed a highly hand-tuned learning rate which involves a warm-up strategy that gradually increases the LR to a larger value and then switching to the regular LR policy (e.g. exponential or polynomial decay). Using LR warm-up and linear scaling, Goyal et al. (2017) managed to train RESNET-50 with batch size 8192 without loss in generalization performance. However, empirical study (Shallue et al., 2018) shows that learning rate scaling heuristics with the batch size do not hold across all problems or across all batch sizes.

More recently, to reduce hand-tuning of hyperparameters, adaptive learning rates for large batch training garnered significant interests (Reddi et al., 2018; Zaheer et al., 2018; Zhang et al., 2019). Several recent works successfully scaled the batch size to large values using adaptive learning rates without degrading the performance, thereby, finishing RESNET-50 training on ImageNet in a few minutes (You et al., 2018; Iandola et al., 2016; Codreanu et al., 2017; Akiba et al., 2017; Jia et al.,

2018; Smith et al., 2017; Martens & Grosse, 2015; Devarakonda et al., 2017; Mikami et al., 2018; Osawa et al., 2018; You et al., 2019; Yamazaki et al., 2019). To the best of our knowledge, the fastest training result for RESNET-50 on ImageNet is due to Ying et al. (2018), who achieve 76+% top-1 accuracy. By using the LARS optimizer and scaling the batch size to 32K on a TPUv3 Pod, Ying et al. (2018) was able to train RESNET-50 on ImageNet in 2.2 minutes. However, it was empirically observed that none of these performance gains hold in other tasks such as BERT training (see Section 4).

## 2 PRELIMINARIES

**Notation.** For any vector $x_t \in \mathbb{R}^d$, either $x_{t,j}$ or $[x_t]_j$ are used to denote its $j^{\text{th}}$ coordinate where $j \in [d]$. Let $\mathbb{I}$ be the $d \times d$ identity matrix, and let $\mathbb{I} = [\mathbb{I}_1, \mathbb{I}_2, ..., \mathbb{I}_h]$ be its decomposition into column submatrices $\mathbb{I}_i = d \times d_h$. For $x \in \mathbb{R}^d$, let $x^{(i)}$ be the block of variables corresponding to the columns of $I_i$ i.e., $x^{(i)} = \mathbb{I}_i^\top x \in \mathbb{R}^{d_i}$ for $i = \{1, 2, \cdots, h\}$. For any function $f : \mathbb{R}^d \to \mathbb{R}$, we use $\nabla_i f(x)$ to denote the gradient with respect to $x^{(i)}$. For any vectors $u, v \in \mathbb{R}^d$, we use $u^2$ and $u/v$ to denote elementwise square and division operators respectively. We use $\|.\|$ and $\|.\|_1$ to denote $l_2$-norm and $l_1$-norm of a vector respectively.

We start our discussion by formally stating the problem setup. In this paper, we study nonconvex stochastic optimization problems of the form

$$\min_{x \in \mathbb{R}^d} f(x) := \mathbb{E}_{s \sim \mathbb{P}}[\ell(x, s)] + \frac{\lambda}{2} \|x\|^2, \tag{1}$$

where $\ell$ is a smooth (possibly nonconvex) function and $\mathbb{P}$ is a probability distribution on the domain $\mathcal{S} \subset \mathbb{R}^k$. Here, $x$ corresponds to model parameters, $\ell$ is the loss function and $\mathbb{P}$ is an unknown data distribution.

We assume function $\ell(x)$ is $L_i$-*smooth* with respect to $i^{\text{th}}$ block, i.e., there exists a constant $L_i$ such that

$$\|\nabla_i \ell(x, s) - \nabla_i \ell(x + I_i \delta, s)\| \le L_i \|\delta\|, \quad \forall \ x \in \mathbb{R}^d, \delta \in \mathbb{R}^{d_i} \text{ and } s \in \mathcal{S}, \tag{2}$$

for all $i \in [h]$. We use $L = (L_1, \cdots, L_h)^\top$ to denote the $h$-dimensional vector of Lipschitz constants. We use $L_\infty$ and $L_{avg}$ to denote $\max_i L_i$ and $\sum_i \frac{L_i}{h}$ respectively. We assume the following bound on the variance in stochastic gradients: $\mathbb{E}\|\nabla_i \ell(x, s) - \nabla_i f(x)\|^2 \le \sigma_i^2$ for all $x \in \mathbb{R}^d$ and $i \in [h]$. Furthermore, we also assume $\mathbb{E}\|[\nabla \ell(x, s)]_i - [\nabla f(x)]_i\|^2 \le \tilde{\sigma}_i^2$ for all $x \in \mathbb{R}^d$ and $i \in [d]$. We use $\sigma = (\sigma_1, \cdots, \sigma_h)^\top$ and $\tilde{\sigma} = (\tilde{\sigma}_1, \cdots, \tilde{\sigma}_d)^\top$ to denote the vectors of standard deviations of stochastic gradient per layer and per dimension respectively. Finally, we assume that the gradients are bounded i.e., $|[\nabla l(x, s)]_j| \le G$ for all $j \in [d]$, $x \in \mathbb{R}^d$ and $s \in \mathcal{S}$. Note that such assumptions are typical in the analysis of stochastic first-order methods (cf. (Ghadimi & Lan, 2013a; Ghadimi et al., 2014; Reddi et al., 2016; 2018)).

Stochastic gradient descent (SGD) is one of the simplest first-order algorithms for solving problem in Equation 1. The update at the $t^{\text{th}}$ iteration of SGD is of the following form:

$$x_{t+1} = x_t - \eta_t \frac{1}{|\mathcal{S}_t|} \sum_{s_t \in \mathcal{S}_t} \nabla \ell(x_t, s_t) + \lambda x_t, \tag{SGD}$$

where $S_t$ is set of $b$ random samples drawn from the distribution $\mathbb{P}$. For very large batch settings, the following is a well-known result for SGD.

**Theorem 1** ((Ghadimi & Lan, 2013b)). *With large batch $b = T$ and using appropriate learning rate, we have the following for the iterates of SGD:*

$$\mathbb{E}\left[\|\nabla f(x_a)\|^2\right] \le O\left(\frac{(f(x_1) - f(x^*))L_\infty}{T} + \frac{\|\sigma\|^2}{T}\right).$$

*where $x^*$ is an optimal solution to the problem in equation 1 and $x_a$ is an iterate uniformly randomly chosen from $\{x_1, \cdots, x_T\}$.*

However, tuning the learning rate $\eta_t$ in SGD, especially in large batch settings, is difficult in practice. Furthermore, the dependence on $L_\infty$ (the maximum of smoothness across dimension) can lead to significantly slow convergence. In the next section, we discuss algorithms to circumvent this issue.

## 3 ALGORITHMS

In this section, we first discuss a general strategy to adapt the learning rate in large batch settings. Using this strategy, we discuss two specific algorithms in the later part of the section. Since our primary focus is on deep learning, our discussion is centered around training a $h$-layer neural network.

**General Strategy.** Suppose we use an iterative *base* algorithm $\mathcal{A}$ (e.g. SGD or ADAM) in the small batch setting with the following layerwise update rule:

$$x_{t+1} = x_t + \eta_t u_t,$$

where $u_t$ is the update made by $\mathcal{A}$ at time step $t$. We propose the following two changes to the update for large batch settings:

1. The update is normalized to unit $l_2$-norm. This is ensured by modifying the update to the form $u_t/\|u_t\|$. Throughout this paper, such a normalization is done layerwise i.e., the update for each layer is ensured to be unit $l_2$-norm.

2. The learning rate is scaled by $\phi(\|x_t\|)$ for some function $\phi : \mathbb{R}^+ \to \mathbb{R}^+$. Similar to the normalization, such a scaling is done layerwise.

Suppose the base algorithm $\mathcal{A}$ is SGD, then the modification results in the following update rule:

$$x_{t+1}^{(i)} = x_t^{(i)} - \eta_t \frac{\phi(\|x_t^{(i)}\|)}{\|g_t^{(i)}\|} g_t^{(i)}, \tag{3}$$

for all layers $i \in [h]$ and where $x_t^{(i)}$ and $g_t^{(i)}$ are the parameters and the gradients of the $i^{\text{th}}$ layer at time step $t$. The normalization modification is similar to one typically used in normalized gradient descent except that it is done layerwise. Note that the modification leads to a biased gradient update; however, in large-batch settings, it can be shown that this bias is small. It is intuitive that such a normalization provides robustness to exploding gradients (where the gradient can be arbitrarily large) and plateaus (where the gradient can be arbitrarily small). Normalization of this form essentially ignores the size of the gradient and is particularly useful in large batch settings where the direction of the gradient is largely preserved.

The scaling term involving $\phi$ ensures that the norm of the update is of the same order as that of the parameter. We found that this typically ensures faster convergence in deep neural networks. In practice, we observed that a simple function of $\phi(z) = \min\{\max\{z, \gamma_l\}, \gamma_u\}$ works well. It is instructive to consider the case where $\phi(z) = z$. In this scenario, the overall change in the learning rate is $\frac{\|x_t^{(i)}\|}{\|g_t^{(i)}\|}$, which can also be interpreted as an estimate on the inverse of Lipschitz constant of the gradient (see equation 2). We now discuss different instantiations of the strategy discussed above. In particular, we focus on two algorithms: LARS (3.1) and the proposed method, LAMB (3.2).

### 3.1 LARS ALGORITHM

The first instantiation of the general strategy is LARS algorithm (You et al., 2017), which is obtained by using momentum optimizer as the base algorithm $\mathcal{A}$ in the framework. LARS was earlier proposed for large batch learning for RESNET on ImageNet. In general, it is observed that the using (heavy-ball) momentum, one can reduce the variance in the stochastic gradients at the cost of little bias. The pseudocode for LARS is provide in Algorithm 1.

We now provide convergence analysis for LARS in general nonconvex setting stated in this paper. For the sake of simplicity, we analyze the case where $\beta_1 = 0$ and $\lambda = 0$ in Algorithm 1. However, our analysis should extend to the general case as well. We will defer all discussions about the convergence rate to the end of the section.

**Theorem 2.** *Let $\eta_t = \eta = \sqrt{\frac{2(f(x_1) - f(x^*))}{\alpha_u^2 \|L\|_1 T}}$ for all $t \in [T]$, $b = T$, $\alpha_l \leq \phi(v) \leq \alpha_u$ for all $v > 0$ where $\alpha_l, \alpha_u > 0$. Then for $x_t$ generated using LARS (Algorithm 1), we have the following bound*

$$\left(\mathbb{E}\left[\frac{1}{\sqrt{h}}\sum_{i=1}^{h}\|\nabla_i f(x_a)\|\right]\right)^2 \leq O\left(\frac{(f(x_1) - f(x^*))L_{avg}}{T} + \frac{\|\sigma\|_1^2}{Th}\right),$$

*where $x^*$ is an optimal solution to the problem in equation 1 and $x_a$ is an iterate uniformly randomly chosen from $\{x_1, \cdots, x_T\}$.*

**Algorithm 1** LARS

**Input:** $x_1 \in \mathbb{R}^d$, learning rate $\{\eta_t\}_{t=1}^T$, parameter $0 < \beta_1 < 1$, scaling function $\phi, \epsilon > 0$
Set $m_0 = 0$
**for** $t = 1$ **to** $T$ **do**
 Draw b samples $S_t$ from $\mathbb{P}$
 Compute $g_t = \frac{1}{|S_t|} \sum_{s_t \in S_t} \nabla \ell(x_t, s_t)$
 $m_t = \beta_1 m_{t-1} + (1 - \beta_1)(g_t + \lambda x_t)$
 $x_{t+1}^{(i)} = x_t^{(i)} - \eta_t \frac{\phi(\|x_t^{(i)}\|)}{\|m_t^{(i)}\|} m_t^{(i)}$ for all $i \in [h]$
**end for**

**Algorithm 2** LAMB

**Input:** $x_1 \in \mathbb{R}^d$, learning rate $\{\eta_t\}_{t=1}^T$, parameters $0 < \beta_1, \beta_2 < 1$, scaling function $\phi, \epsilon > 0$
Set $m_0 = 0, v_0 = 0$
**for** $t = 1$ **to** $T$ **do**
 Draw b samples $S_t$ from $\mathbb{P}$.
 Compute $g_t = \frac{1}{|S_t|} \sum_{s_t \in S_t} \nabla \ell(x_t, s_t)$.
 $m_t = \beta_1 m_{t-1} + (1 - \beta_1)g_t$
 $v_t = \beta_2 v_{t-1} + (1 - \beta_2)g_t^2$
 $m_t = m_t/(1 - \beta_1^t)$
 $v_t = v_t/(1 - \beta_2^t)$
 Compute ratio $r_t = \frac{m_t}{\sqrt{v_t} + \epsilon}$
 $x_{t+1}^{(i)} = x_t^{(i)} - \eta_t \frac{\phi(\|x_t^{(i)}\|)}{\|r_t^{(i)} + \lambda x_t^{(i)}\|}(r_t^{(i)} + \lambda x_t^{(i)})$
**end for**

## 3.2 LAMB ALGORITHM

The second instantiation of the general strategy is obtained by using ADAM as the base algorithm $\mathcal{A}$. ADAM optimizer is popular in deep learning community and has shown to have good performance for training state-of-the-art language models like BERT. Unlike LARS, the adaptivity of LAMB is two-fold: (i) per dimension normalization with respect to the square root of the second moment used in ADAM and (ii) layerwise normalization obtained due to layerwise adaptivity. The pseudocode for LAMB is provided in Algorithm 2. When $\beta_1 = 0$ and $\beta_2 = 0$, the algorithm reduces to be Sign SGD where the learning rate is scaled by square root of the layer dimension (Bernstein et al., 2018).

The following result provides convergence rate for LAMB in general nonconvex settings. Similar to the previous case, we focus on the setting where $\beta_1 = 0$ and $\lambda = 0$. As before, our analysis extends to the general case; however, the calculations become messy.

**Theorem 3.** *Let $\eta_t = \eta = \sqrt{\frac{2(f(x_1) - f(x^*))}{\alpha_u^2 \|L\|_1 T}}$ for all $t \in [T]$, $b = T$, $d_i = d/h$ for all $i \in [h]$, and $\alpha_l \le \phi(v) \le \alpha_u$ for all $v > 0$ where $\alpha_l, \alpha_u > 0$. Then for $x_t$ generated using LAMB (Algorithm 2), we have the following bounds:*

 *1. When $\beta_2 = 0$, we have*

$$\left(\mathbb{E}\left[\frac{1}{\sqrt{d}}\|\nabla f(x_a)\|_1\right]\right)^2 \le O\left(\frac{(f(x_1) - f(x^*))L_{avg}}{T} + \frac{\|\tilde{\sigma}\|_1^2}{Th}\right),$$

 *2. When $\beta_2 > 0$, we have*

$$\mathbb{E}[\|\nabla f(x_a)\|^2] \le O\left(\sqrt{\frac{G^2 d}{h(1 - \beta_2)}} \times \left[\sqrt{\frac{2(f(x_1) - f(x^*))\|L\|_1}{T}} + \frac{\|\tilde{\sigma}\|_1}{\sqrt{T}}\right]\right),$$

*where $x^*$ is an optimal solution to the problem in equation 1 and $x_a$ is an iterate uniformly randomly chosen from $\{x_1, \cdots, x_T\}$.*

**Discussion on convergence rates.** We first start our discussion with the comparison of convergence rate of LARS with that of SGD (Theorem 1). The convergence rates of LARS and SGD differ in two ways: (1) the convergence criterion is $(\mathbb{E}[\sum_{i=1}^h \|\nabla_i f\|])^2$ as opposed to $\mathbb{E}[\|\nabla f\|^2]$ in SGD and (2) the dependence on $L$ and $\sigma$ in the convergence rate. Briefly, the convergence rate of LARS is better than SGD when the gradient is denser than curvature and stochasticity. This convergence rate comparison is similar in spirit to the one obtained in (Bernstein et al., 2018). Assuming that the convergence criterion in Theorem 1 and Theorem 2 is of similar order (which happens when gradients are fairly dense), convergence rate of LARS and LAMB depend on $L_{avg}$ instead of $L_\infty$ and are thus, significantly better than that of SGD. A more quantitative comparison is provided in Section C of the Appendix. The comparison of LAMB (with $\beta_2 = 0$) with SGD is along similar lines. We obtain slightly worse rates for the case where $\beta_2 > 0$; although, we believe that its behavior should be better than the case $\beta_2 = 0$. We leave this investigation to future work.

# 4 EXPERIMENTS

We now present empirical results comparing LAMB with existing optimizers on two important large batch training tasks: BERT and RESNET-50 training. We also compare LAMB with existing optimizers for small batch size ($< 1K$) and small dataset (e.g. CIFAR, MNIST) (see Appendix).

**Experimental Setup.** To demonstrate its robustness, we use very minimal hyperparameter tuning for the LAMB optimizer. Thus, it is possible to achieve better results by further tuning the hyperparameters. The parameters $\beta_1$ and $\beta_2$ in Algorithm 2 are set to 0.9 and 0.999 respectively in all our experiments; we only tune the learning rate. We use a polynomially decaying learning rate of $\eta_t = \eta_0 \times (1 - t/T)$ in Algorithm 2), which is the same as in BERT baseline. This setting also works for all other applications in this paper. Furthermore, for BERT and RESNET-50 training, we did not tune the hyperparameters of LAMB while increasing the batch size. We use the square root of LR scaling rule to automatically adjust learning rate and linear-epoch warmup scheduling. We use TPUv3 in all the experiments. A TPUv3 Pod has 1024 chips and can provide more than 100 petaflops performance for mixed precision computing. To make sure we are comparing with solid baselines, we use grid search to tune the hyper-parameters for ADAM, ADAGRAD, ADAMW (ADAM with weight decay), and LARS. We also tune weight decay for ADAMW. All the hyperparameter tuning settings are reported in the Appendix. Due to space constraints, several experimental details are relegated to the Appendix.

## 4.1 BERT TRAINING

We first discuss empirical results for speeding up BERT training. For this experiment, we use the same dataset as Devlin et al. (2018), which is a concatenation of Wikipedia and BooksCorpus with 2.5B and 800M words respectively. We specifically focus on the SQuAD task[2] in this paper. The F1 score on SQuAD-v1 is used as the accuracy metric in our experiments. All our comparisons are with respect to the baseline BERT model by Devlin et al. (2018). To train BERT, Devlin et al. (2018) first train the model for 900k iterations using a sequence length of 128 and then switch to a sequence length of 512 for the last 100k iterations. This results in a training time of around 3 days on 16 TPUv3 chips. The baseline BERT model[3] achieves a F1 score of 90.395. To ensure a fair comparison, we follow the same SQuAD fine-tune procedure of Devlin et al. (2018) without modifying any configuration (including number of epochs and hyperparameters). As noted earlier, we could get even better results by changing the fine-tune configuration. For instance, by just slightly changing the learning rate in the fine-tune stage, we can obtain a higher F1 score of 91.688 for the batch size of 16K using LAMB. We report a F1 score of 91.345 in Table 1, which is the score obtained for the untuned version. Below we describe two different training choices for training BERT and discuss the corresponding speedups.

For the first choice, we maintain the same training procedure as the baseline except for changing the training optimizer to LAMB. We run with the same number of epochs as the baseline but with batch size scaled from 512 to 32K. The choice of 32K batch size (with sequence length 512) is mainly due to memory limits of TPU Pod. Our results are shown in Table 1. By using the LAMB optimizer, we are able to achieve a F1 score of 91.460 in 15625 iterations for a batch size of 32768 (14063 iterations for sequence length 128 and 1562 iterations for sequence length 512). With 32K batch size, we reduce BERT training time from 3 days to around 100 minutes. We achieved 49.1 times speedup by 64 times computational resources (76.7% efficiency). We consider the speedup is great because we use the synchronous data-parallelism. There is a communication overhead coming from transferring of the gradients over the interconnect. For RESNET-50, researchers are able to achieve 90% scaling efficiency because RESNET-50 has much fewer parameters (# parameters is equal to #gradients) than BERT (25 million versus 300 million).

To obtain further improvements, we use the **Mixed-Batch Training** procedure with LAMB. Recall that BERT training involves two stages: the first 9/10 of the total epochs use a sequence length of 128, while the last 1/10 of the total epochs use a sequence length of 512. For the second stage training, which involves a longer sequence length, due to memory limits, a maximum batch size of only 32768 can be used on a TPUv3 Pod. However, we can potentially use a larger batch size for the first stage because of a shorter sequence length. In particular, the batch size can be increased to 131072 for the first stage. However, we did not observe any speedup by increasing the batch size from 65536 to 131072 for the first stage, thus, we restrict the batch size to 65536 for this stage. By using this strategy, we are able to make full utilization of the hardware resources throughout the training

---

[2]https://rajpurkar.github.io/SQuAD-explorer/
[3]Pre-trained BERT model can be downloaded from https://github.com/google-research/bert

Table 1: We use the F1 score on SQuAD-v1 as the accuracy metric. The baseline F1 score is the score obtained by the pre-trained model (BERT-Large) provided on BERT's public repository (as of February 1st, 2019). We use TPUv3s in our experiments. We use the same setting as the baseline: the first 9/10 of the total epochs used a sequence length of 128 and the last 1/10 of the total epochs used a sequence length of 512. All the experiments run the same number of epochs. Dev set means the test data. It is worth noting that we can achieve better results by manually tuning the hyperparameters. The data in this table is collected from the untuned version.

| Solver | batch size | steps | F1 score on dev set | TPUs | Time |
|--------|-----------|-------|---------------------|------|------|
| Baseline | 512 | 1000k | 90.395 | 16 | 81.4h |
| LAMB | 512 | 1000k | 91.752 | 16 | 82.8h |
| LAMB | 1k | 500k | 91.761 | 32 | 43.2h |
| LAMB | 2k | 250k | 91.946 | 64 | 21.4h |
| LAMB | 4k | 125k | 91.137 | 128 | 693.6m |
| LAMB | 8k | 62500 | 91.263 | 256 | 390.5m |
| LAMB | 16k | 31250 | 91.345 | 512 | 200.0m |
| LAMB | 32k | 15625 | 91.475 | 1024 | 101.2m |
| LAMB | 64k/32k | 8599 | 90.584 | 1024 | 76.19m |

procedure. Increasing the batch size is able to warm-up and stabilize the optimization process (Smith et al., 2017), but decreasing the batch size brings chaos to the optimization process and can cause divergence. In our experiments, we found a technique that is useful to stabilize the second stage optimization. Because we switched to a different optimization problem, it is necessary to re-warm-up the optimization. Instead of decaying the learning rate at the second stage, we ramp up the learning rate from zero again in the second stage (re-warm-up). As with the first stage, we decay the learning rate after the re-warm-up phase. With this method, we only need 8599 iterations and finish BERT training in 76 minutes (100.2% efficiency).

**Comparison with ADAMW and LARS.** To ensure that our approach is compared to a solid baseline for the BERT training, we tried three different strategies for tuning ADAMW (Loshchilov & Hutter, 2017): (1) ADAMW with default hyperparameters (Devlin et al., 2018) (2) ADAMW with the same hyperparameters as LAMB, and (3) ADAMW with tuned hyperparameters. ADAMW stops scaling at the batch size of 16K because it is not able to achieve the target F1 score (88.1 vs 90.4). The tuning information of ADAMW is shown in the Appendix. For 64K/32K mixed-batch training, even after extensive tuning of the hyperparameters, we fail to get any reasonable result with ADAMW optimizer. We conclude that ADAMW does not work well in large-batch BERT training or is at least hard to tune. We also observe that LAMB performs better than LARS for all batch sizes (Table 2).

Table 2: LAMB achieves a higher performance (F1 score) than LARS for all the batch sizes. The baseline achieves a F1 score of 90.390. Thus, LARS stops scaling at the batch size of 16K.

| Batch Size | 512 | 1K | 2K | 4K | 8K | 16K | 32K |
|-----------|-----|-----|-----|-----|-----|------|------|
| LARS | 90.717 | 90.369 | 90.748 | 90.537 | 90.548 | 89.589 | diverge |
| LAMB | 91.752 | 91.761 | 91.946 | 91.137 | 91.263 | 91.345 | 91.475 |

## 4.2 IMAGENET TRAINING WITH RESNET-50.

ImageNet training with ResNet-50 is an industry standard metric that is being used in MLPerf[4]. The baseline can get 76.3% top-1 accuracy in 90 epochs (Goyal et al., 2017). All the successful implementations are based on momentum SGD (He et al., 2016; Goyal et al., 2017) or LARS optimizer (Ying et al., 2018; Jia et al., 2018; Mikami et al., 2018; You et al., 2018; Yamazaki et al., 2019). Before our study, we did not find any paper reporting a state-of-the-art accuracy achieved

---

[4]https://mlperf.org/

by ADAM (Kingma & Ba, 2014), ADAGRAD, or ADAMW optimizer. In our experiments, even with comprehensive hyper-parameter tuning, ADAGRAD/ADAM/ADAMW (with batch size 16K) only achieves 55.38%/66.04%/67.27% top-1 accuracy. After adding learning rate scheme of Goyal et al. (2017), the top-1 accuracy of ADAGRAD/ADAM/ADAMW was improved to 72.0%/73.48%/73.07%. However, they are still much lower than 76.3%. The details of the tuning information are in the Appendix. Table 3 shows that LAMB can achieve the target accuracy. Beyond a batch size of 8K, LAMB's accuracy is higher than the momentum. LAMB's accuracy is also slightly better than LARS. At a batch size of 32K, LAMB achieves 76.4% top-1 accuracy while LARS achieves 76.3%. At a batch size of 2K, LAMB is able to achieve 77.11% top-1 accuracy while LARS achieves 76.6%.

Table 3: Top-1 validation accuracy of ImageNet/RESNET-50 training at the batch size of 16K (90 epochs). The performance of momentum was reported by (Goyal et al., 2017). + means adding the learning rate scheme of Goyal et al. (2017) to the optimizer: (1) 5-epoch warmup to stablize the initial stage; and (2) multiply the learning rate by 0.1 at 30th, 60th, and 80th epoch. The target accuracy is around 0.763 (Goyal et al., 2017). All the adaptive solvers were comprehensively tuned. The tuning information was in the Appendix.

| optimizer | adagrad/adagrad+ | adam/adam+ | adamw/adamw+ | momentum | lamb |
|---|---|---|---|---|---|
| Accuracy | 0.5538/0.7201 | 0.6604/0.7348 | 0.6727/0.7307 | 0.7520 | 0.7666 |

### 4.3 HYPERPARAMETERS FOR SCALING THE BATCH SIZE

For BERT and ImageNet training, we did not tune the hyperparameters of LAMB optimizer when increasing the batch size. We use the square root LR scaling rule and linear-epoch warmup scheduling to automatically adjust learning rate. The details can be found in Tables 4 and 5

Table 4: Untuned LAMB for BERT training across different batch sizes (fixed #epochs). We use square root LR scaling and linear-epoch warmup. For example, batch size 32K needs to finish 15625 iterations. It uses $0.2 \times 15625 = 3125$ iterations for learning rate warmup. BERT's baseline achieved a F1 score of 90.395. We can achieve an even higher F1 score if we manually tune the hyperparameters.

| Batch Size | 512 | 1K | 2K | 4K | 8K | 16K | 32K |
|---|---|---|---|---|---|---|---|
| Learning Rate | $\frac{5}{2^{3.0} \times 10^3}$ | $\frac{5}{2^{2.5} \times 10^3}$ | $\frac{5}{2^{2.0} \times 10^3}$ | $\frac{5}{2^{1.5} \times 10^3}$ | $\frac{5}{2^{1.0} \times 10^3}$ | $\frac{5}{2^{0.5} \times 10^3}$ | $\frac{5}{2^{0.0} \times 10^3}$ |
| Warmup Ratio | $\frac{1}{320}$ | $\frac{1}{160}$ | $\frac{1}{80}$ | $\frac{1}{40}$ | $\frac{1}{20}$ | $\frac{1}{10}$ | $\frac{1}{5}$ |
| F1 score | 91.752 | 91.761 | 91.946 | 91.137 | 91.263 | 91.345 | 91.475 |
| Exact Match | 85.090 | 85.260 | 85.355 | 84.172 | 84.901 | 84.816 | 84.939 |

Table 5: Untuned LAMB for ImageNet training with RESNET-50 for different batch sizes (90 epochs). We use square root LR scaling and linear-epoch warmup. The baseline Goyal et al. (2017) gets 76.3% top-1 accuracy in 90 epochs. Stanford DAWN Bench (Coleman et al., 2017) baseline achieves 93% top-5 accuracy. LAMB achieves both of them. LAMB can achieve an even higher accuracy if we manually tune the hyperparameters.

| Batch Size | 512 | 1K | 2K | 4K | 8K | 16K | 32K |
|---|---|---|---|---|---|---|---|
| Learning Rate | $\frac{4}{2^{3.0} \times 100}$ | $\frac{4}{2^{2.5} \times 100}$ | $\frac{4}{2^{2.0} \times 100}$ | $\frac{4}{2^{1.5} \times 100}$ | $\frac{4}{2^{1.0} \times 100}$ | $\frac{4}{2^{0.5} \times 100}$ | $\frac{4}{2^{0.0} \times 100}$ |
| Warmup Epochs | 0.3125 | 0.625 | 1.25 | 2.5 | 5 | 10 | 20 |
| Top-5 Accuracy | 0.9335 | 0.9349 | 0.9353 | 0.9332 | 0.9331 | 0.9322 | 0.9308 |
| Top-1 Accuracy | 0.7696 | 0.7706 | 0.7711 | 0.7692 | 0.7689 | 0.7666 | 0.7642 |

## 5 CONCLUSION

Large batch techniques are critical to speeding up deep neural network training. In this paper, we propose the LAMB optimizer, which supports adaptive elementwise updating and layerwise learning

rates. Furthermore, LAMB is a general purpose optimizer that works for both small and large batches. We also provided theoretical analysis for the LAMB optimizer, highlighting the cases where it performs better than standard SGD. LAMB achieves a better performance than existing optimizers for a wide range of applications. By using LAMB, we are able to scale the batch size of BERT pre-training to 64K without losing accuracy, thereby, reducing the BERT training time from 3 days to around 76 minutes. LAMB is also the first large batch adaptive solver that can achieve state-of-the-art accuracy on ImageNet training with RESNET-50.

## 6 ACKNOWLEDGEMENT

We want to thank the comments from George Dahl and Jeff Dean. We want to thank Michael Banfield, Dehao Chen, Youlong Cheng, Sameer Kumar, and Zak Stone for TPU Pod support.

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

APPENDIX

## A  PROOF OF THEOREM 2

*Proof.* We analyze the convergence of LARS for general minibatch size here. Recall that the update of LARS is the following

$$x_{t+1}^{(i)} = x_t^{(i)} - \eta_t \phi(\|x_t^{(i)}\|) \frac{g_t^{(i)}}{\|g_t^{(i)}\|},$$

for all $i \in [h]$. Since the function $f$ is $L$-smooth, we have the following:

$$f(x_{t+1}) \leq f(x_t) + \langle \nabla_i f(x_t), x_{t+1}^{(i)} - x_t^{(i)} \rangle + \sum_{i=1}^{h} \frac{L_i}{2} \|x_{t+1}^{(i)} - x_t^{(i)}\|^2$$

$$= f(x_t) - \eta_t \sum_{i=1}^{h} \sum_{j=1}^{d_i} \phi(\|x_t^{(i)}\|) \times \left( [\nabla_i f(x_t)]_j \times \frac{g_{t,j}^{(i)}}{\|g_t^{(i)}\|} \right) + \sum_{i=1}^{h} \frac{L_i \eta_t^2 \phi^2(\|x_t^{(i)}\|)}{2}$$

$$\leq f(x_t) - \eta_t \sum_{i=1}^{h} \sum_{j=1}^{d_i} \phi(\|x_t^{(i)}\|) \times \left( [\nabla_i f(x_t)]_j \times \left( \frac{g_{t,j}^{(i)}}{\|g_t^{(i)}\|} - \frac{[\nabla_i f(x_t)]_j}{\|\nabla_i f(x_t)\|} + \frac{[\nabla_i f(x_t)]_j}{\|\nabla_i f(x_t)\|} \right) \right) + \frac{\eta_t^2 \alpha_u^2}{2} \|L\|_1$$

$$= f(x_t) - \eta_t \sum_{i=1}^{h} \phi(\|x_t^{(i)}\|) \times \|\nabla_i f(x_t)\|$$

$$- \eta_t \sum_{i=1}^{h} \sum_{j=1}^{d_i} \phi(\|x_t^{(i)}\|) \times \left( [\nabla_i f(x_t)]_j \times \left( \frac{g_{t,j}^{(i)}}{\|g_t^{(i)}\|} - \frac{[\nabla_i f(x_t)]_j}{\|\nabla_i f(x_t)\|} \right) \right) + \frac{\eta_t^2 \alpha_u^2}{2} \|L\|_1$$

$$\tag{4}$$

The first inequality follows from the lipschitz continuous nature of the gradient. Let $\Delta_t^{(i)} = g_t^{(i)} - \nabla_i f(x_t)$. Then the above inequality can be rewritten in the following manner:

$$f(x_{t+1}) \leq f(x_t) - \eta_t \sum_{i=1}^{h} \phi(\|x_t^{(i)}\|) \|\nabla_i f(x_t)\|$$

$$- \eta_t \sum_{i=1}^{h} \sum_{j=1}^{d_i} \phi(\|x_t^{(i)}\|) \times \left( [\nabla_i f(x_t)]_j \times \left( \frac{(\Delta_{t,j}^{(i)} + [\nabla_i f(x_t)]_j)}{\|\Delta_t^{(i)} + \nabla_i f(x_t)\|} - \frac{[\nabla_i f(x_t)]_j}{\|\nabla_i f(x_t)\|} \right) \right) + \frac{\eta_t^2 \alpha_u^2}{2} \|L\|_1$$

$$= f(x_t) - \eta_t \sum_{i=1}^{h} \phi(\|x_t^{(i)}\|) \|\nabla_i f(x_t)\|$$

$$- \eta_t \sum_{i=1}^{h} \phi(\|x_t^{(i)}\|) \times \left( \frac{\langle \Delta_t^{(i)} + \nabla_i f(x_t), \nabla_i f(x_t) \rangle}{\|\Delta_t^{(i)} + \nabla_i f(x_t)\|} - \|\nabla_i f(x_t)\| \right) + \frac{\eta_t^2 \alpha_u^2}{2} \|L\|_1$$

$$= f(x_t) - \eta_t \sum_{i=1}^{h} \phi(\|x_t^{(i)}\|) \|\nabla_i f(x_t)\|$$

$$+ \eta_t \sum_{i=1}^{h} \phi(\|x_t^{(i)}\|) \times \left( \frac{\|\nabla_i f(x_t)\| \|\Delta_t^{(i)} + \nabla_i f(x_t)\| - \langle \Delta_t^{(i)} + \nabla_i f(x_t), \nabla_i f(x_t) \rangle}{\|\Delta_t^{(i)} + \nabla_i f(x_t)\|} \right) + \frac{\eta_t^2 \alpha_u^2}{2} \|L\|_1$$

$$= f(x_t) - \eta_t \sum_{i=1}^{h} \phi(\|x_t^{(i)}\|) \|\nabla_i f(x_t)\| + \frac{\eta_t^2 \alpha_u^2}{2} \|L\|_1$$

$$+ \eta_t \sum_{i=1}^{h} \phi(\|x_t^{(i)}\|) \times \left( \frac{\|\nabla_i f(x_t)\| \|\Delta_t^{(i)} + \nabla_i f(x_t)\| - \|\Delta_t^{(i)} + \nabla_i f(x_t)\|^2 + \langle \Delta_t^{(i)}, \Delta_t^{(i)} + \nabla_i f(x_t) \rangle}{\|\Delta_t^{(i)} + \nabla_i f(x_t)\|} \right).$$

$$\tag{5}$$

Using Cauchy-Schwarz inequality in the above inequality, we have:

$$f(x_{t+1}) \leq f(x_t) - \eta_t \sum_{i=1}^{h} \phi(\|x_t^{(i)}\|)\|\nabla_i f(x_t)\|$$

$$+ \eta_t \sum_{i=1}^{h} \phi(\|x_t^{(i)}\|) \times \left( \|\nabla_i f(x_t)\| - \|\Delta_t^{(i)} + \nabla_i f(x_t)\| + \|\Delta_t^{(i)}\| \right) + \frac{\eta_t^2 \alpha_u^2}{2}\|L\|_1$$

$$\leq f(x_t) - \eta_t \sum_{i=1}^{h} \phi(\|x_t^{(i)}\|)\|\nabla_i f(x_t)\| + 2\eta_t \sum_{i=1}^{h} \phi(\|x_t^{(i)}\|) \times \|\Delta_t^{(i)}\| + \frac{\eta_t^2 \alpha_u^2}{2}\|L\|_1$$

Taking expectation, we obtain the following:

$$\mathbb{E}[f(x_{t+1})] \leq f(x_t) - \eta_t \sum_{i=1}^{h} \phi(\|x_t^{(i)}\|)\|\nabla_i f(x_t)\| + 2\eta_t \sum_{i=1}^{h} \phi(\|x_t^{(i)}\|) \times \mathbb{E}[\|\Delta_t^{(i)}\|] + \frac{\eta_t^2 \alpha_u^2}{2}\|L\|_1$$

$$\leq f(x_t) - \eta_t \alpha_l \sum_{i=1}^{h} \|\nabla_i f(x_t)\| + 2\eta_t \alpha_u \frac{\|\sigma\|_1}{\sqrt{b}} + \frac{\eta_t^2 \alpha_u^2}{2}\|L\|_1. \tag{6}$$

Summing the above inequality for $t = 1$ to $T$ and using telescoping sum, we have the following inequality:

$$\mathbb{E}[f(x_{T+1})] \leq f(x_1) - \eta\alpha_l \sum_{t=1}^{T}\sum_{i=1}^{h} \mathbb{E}[\|\nabla_i f(x_t)\|] + 2\eta T \frac{\alpha_u\|\sigma\|_1}{\sqrt{b}} + \frac{\eta^2 \alpha_u^2 T}{2}\|L\|_1.$$

Rearranging the terms of the above inequality, and dividing by $\eta T \alpha_l$, we have:

$$\frac{1}{T}\sum_{t=1}^{T}\sum_{i=1}^{h} \mathbb{E}[\|\nabla_i f(x_t)\|] \leq \frac{f(x_1) - \mathbb{E}[f(x_{T+1})]}{T\eta\alpha_l} + \frac{2\alpha_u\|\sigma\|_1}{\sqrt{b}\alpha_l} + \frac{\eta\alpha_u^2}{2\alpha_l}\|L\|_1$$

$$\leq \frac{f(x_1) - f(x^*)}{T\eta\alpha_l} + \frac{2\alpha_u\|\sigma\|_1}{\alpha_l\sqrt{b}} + \frac{\eta\alpha_u^2}{2\alpha_l}\|L\|_1.$$

$\square$

## B  PROOF OF THEOREM 3

*Proof.* We analyze the convergence of LAMB for general minibatch size here. Recall that the update of LAMB is the following

$$x_{t+1}^{(i)} = x_t^{(i)} - \eta_t \phi(\|x_t^{(i)}\|)\frac{r_t^{(i)}}{\|r_t^{(i)}\|},$$

for all $i \in [h]$. For simplicity of notation, we reason the

Since the function $f$ is $L$-smooth, we have the following:

$$f(x_{t+1}) \leq f(x_t) + \langle \nabla_i f(x_t), x_{t+1}^{(i)} - x_t^{(i)}\rangle + \sum_{i=1}^{h} \frac{L_i}{2}\|x_{t+1}^{(i)} - x_t^{(i)}\|^2$$

$$= f(x_t) - \eta_t \underbrace{\sum_{i=1}^{h}\sum_{j=1}^{d_i} \phi(\|x_t^{(i)}\|) \times \left( [\nabla_i f(x_t)]_j \times \frac{r_{t,j}^{(i)}}{\|r_t^{(i)}\|} \right)}_{T_1} + \sum_{i=1}^{h} \frac{L_i \alpha_u^2 \eta_t^2}{2} \tag{7}$$

The above inequality simply follows from the lipschitz continuous nature of the gradient. We bound term $T_1$ in the following manner:

$$T_1 \leq -\eta_t \sum_{i=1}^{h} \sum_{j=1}^{d_i} \phi(\|x_t^{(i)}\|) \times \left( [\nabla_i f(x_t)]_j \times \frac{r_{t,j}^{(i)}}{\|r_t^{(i)}\|} \right)$$

$$\leq -\eta_t \sum_{i=1}^{h} \sum_{j=1}^{d_i} \sqrt{\frac{1-\beta_2}{G^2 d_i}} \left( \phi(\|x_t^{(i)}\|) \times [\nabla_i f(x_t)]_j \times g_{t,j}^{(i)} \right)$$

$$- \eta_t \sum_{i=1}^{h} \sum_{j=1}^{d_i} \left( \phi(\|x_t^{(i)}\|) \times [\nabla_i f(x_t)]_j \times \frac{r_{t,j}^{(i)}}{\|r_t^{(i)}\|} \right) \mathbb{1}(sign(\nabla_i f(x_t)]_j) \neq sign(r_{t,j}^{(i)}))$$

(8)

This follows from the fact that $\|r_t^{(i)}\| \leq \sqrt{\frac{d_i}{1-\beta_2}}$ and $\sqrt{v_t} \leq G$. If $\beta_2 = 0$, then $T_1$ can be bounded as follows:

$$T_1 \leq -\eta_t \sum_{i=1}^{h} \sum_{j=1}^{d_i} \sqrt{\frac{1}{d_i}} \left( \phi(\|x_t^{(i)}\|) \times |[\nabla_i f(x_t)]_j| \right)$$

$$- \eta_t \sum_{i=1}^{h} \sum_{j=1}^{d_i} \left( \phi(\|x_t^{(i)}\|) \times [\nabla_i f(x_t)]_j \times \frac{r_{t,j}^{(i)}}{\|r_t^{(i)}\|} \right) \mathbb{1}(sign(\nabla_i f(x_t)]_j) \neq sign(r_{t,j}^{(i)}))$$

The rest of the proof for $\beta_2 = 0$ is similar to argument for the case $\beta_2 > 0$, which is shown below. Taking expectation, we have the following:

$$\mathbb{E}[T_1] \leq -\eta_t \sum_{i=1}^{h} \sum_{j=1}^{d_i} \sqrt{\frac{1-\beta_2}{G^2 d_i}} \mathbb{E}\left[ \phi(\|x_t^{(i)}\|) \times \left( [\nabla_i f(x_t)]_j \times g_{t,j}^{(i)} \right) \right]$$

$$- \eta_t \sum_{i=1}^{h} \sum_{j=1}^{d_i} \mathbb{E}\left[ \phi(\|x_t^{(i)}\|) \times \left( [\nabla_i f(x_t)]_j \times \frac{r_{t,j}^{(i)}}{\|r_t^{(i)}\|} \right) \mathbb{1}(sign(\nabla_i f(x_t)]_j) \neq sign(g_{t,j}^{(i)})) \right]$$

$$\leq -\eta_t \sum_{i=1}^{h} \sum_{j=1}^{d_i} \sqrt{\frac{1-\beta_2}{G^2 d_i}} \mathbb{E}\left[ \left( \phi(\|x_t^{(i)}\|) \times [\nabla_i f(x_t)]_j \times g_{t,j}^{(i)} \right) \right]$$

$$+ \eta_t \sum_{i=1}^{h} \sum_{j=1}^{d_i} \mathbb{E}\left[ \alpha_u |[\nabla_i f(x_t)]_j| \mathbb{1}(sign(\nabla_i f(x_t)]_j) \neq sign(g_{t,j}^{(i)})) \right]$$

$$\leq -\eta_t \sum_{i=1}^{h} \sum_{j=1}^{d_i} \sqrt{\frac{1-\beta_2}{G^2 d_i}} \mathbb{E}\left[ \phi(\|x_t^{(i)}\|) \times \left( [\nabla_i f(x_t)]_j \times g_{t,j}^{(i)} \right) \right]$$

$$+ \eta_t \sum_{i=1}^{h} \sum_{j=1}^{d_i} \alpha_u |[\nabla_i f(x_t)]_j| \mathbb{P}(sign(\nabla_i f(x_t)]_j) \neq sign(g_{t,j}^{(i)}))$$

Using the bound on the probability that the signs differ, we get:

$$\mathbb{E}[T_1] \leq -\eta_t \alpha_l \sqrt{\frac{h(1-\beta_2)}{G^2 d}} \|\nabla f(x_t)\|^2 + \eta_t \alpha_u \sum_{i=1}^{h} \sum_{j=1}^{d_i} \frac{\sigma_{i,j}}{\sqrt{b}}.$$

Substituting the above bound on $T_1$ in equation 7, we have the following bound:

$$\mathbb{E}[f(x_{t+1})] \leq f(x_t) - \eta_t \alpha_l \sqrt{\frac{h(1-\beta_2)}{G^2 d}} \|\nabla f(x_t)\|^2 + \eta_t \alpha_u \frac{\|\tilde{\sigma}\|_1}{\sqrt{b}} + \frac{\eta_t^2 \alpha_u^2 \|L\|_1}{2}$$

(9)

Summing the above inequality for $t = 1$ to $T$ and using telescoping sum, we have the following inequality:

$$\mathbb{E}[f(x_{T+1})] \leq f(x_1) - \eta_t \alpha_l \sqrt{\frac{h(1-\beta_2)}{G^2 d}} \sum_{t=1}^{T} \mathbb{E}[\|\nabla f(x_t)\|^2] + \eta T \alpha_u \frac{\|\tilde{\sigma}\|_1}{\sqrt{b}} + \frac{\eta^2 \alpha_u^2 T}{2} \|L\|_1.$$

**Algorithm 3** N-LAMB

**Input:** $x_1 \in \mathbb{R}^d$, learning rate $\{\eta_t\}_{t=1}^T$, parameters $0 < \beta_1, \beta_2 < 1$, scaling function $\phi$, $\epsilon > 0$, parameters $0 < \{\beta_1^t\}_{t=1}^T < 1$
Set $m_0 = 0$, $v_0 = 0$
**for** $t = 1$ to $T$ **do**
    Draw b samples $S_t$ from $\mathbb{P}$.
    Compute $g_t = \frac{1}{|S_t|} \sum_{s_t \in S_t} \nabla \ell(x_t, s_t)$.
    $m_t = \beta_1 m_{t-1} + (1 - \beta_1) g_t$
    $\hat{m} = \frac{\beta_1^{t+1} m_t}{1 - \Pi_{i=1}^{t+1} \beta_1^i} + \frac{(1-\beta_1) g_t}{1 - \Pi_{i=1}^t \beta_1^i}$
    $v_t = \beta_2 v_{t-1} + (1 - \beta_2) g_t^2$
    $\hat{v} = \frac{\beta_2 v_t}{1 - \beta_2^t}$
    Compute ratio $r_t = \frac{\hat{m}}{\sqrt{\hat{v}} + \epsilon}$
    $x_{t+1}^{(i)} = x_t^{(i)} - \eta_t \frac{\phi(\|x_t^{(i)}\|)}{\|r_t^{(i)} + \lambda x_t^{(i)}\|} (r_t^{(i)} + \lambda x_t)$
**end for**

**Algorithm 4** NN-LAMB

**Input:** $x_1 \in \mathbb{R}^d$, learning rate $\{\eta_t\}_{t=1}^T$, parameters $0 < \beta_1, \beta_2 < 1$, scaling function $\phi$, $\epsilon > 0$, parameters $0 < \{\beta_1^t\}_{t=1}^T < 1$
Set $m_0 = 0$, $v_0 = 0$
**for** $t = 1$ to $T$ **do**
    Draw b samples $S_t$ from $\mathbb{P}$.
    Compute $g_t = \frac{1}{|S_t|} \sum_{s_t \in S_t} \nabla \ell(x_t, s_t)$.
    $m_t = \beta_1 m_{t-1} + (1 - \beta_1) g_t$
    $\hat{m} = \frac{\beta_1^{t+1} m_t}{1 - \Pi_{i=1}^{t+1} \beta_1^i} + \frac{(1-\beta_1^t) g_t}{1 - \Pi_{i=1}^t \beta_1^i}$
    $v_t = \beta_2 v_{t-1} + (1 - \beta_2) g_t^2$
    $\hat{v} = \frac{\beta_2^{t+1} v_t}{1 - \Pi_{i=1}^{t+1} \beta_2^i} + \frac{(1-\beta_2^t) g_t^2}{1 - \Pi_{i=1}^t \beta_2^i}$
    Compute ratio $r_t = \frac{\hat{m}}{\sqrt{\hat{v}} + \epsilon}$
    $x_{t+1}^{(i)} = x_t^{(i)} - \eta_t \frac{\phi(\|x_t^{(i)}\|)}{\|r_t^{(i)} + \lambda x_t^{(i)}\|} (r_t^{(i)} + \lambda x_t)$
**end for**

Rearranging the terms of the above inequality, and dividing by $\eta T \alpha_l$, we have:

$$\sqrt{\frac{h(1-\beta_2)}{G^2 d}} \frac{1}{T} \sum_{t=1}^T \mathbb{E}[\|\nabla f(x_t)\|^2] \leq \frac{f(x_1) - \mathbb{E}[f(x_{T+1})]}{T\eta\alpha_l} + \frac{\alpha_u \|\tilde{\sigma}\|_1}{\alpha_l \sqrt{b}} + \frac{\eta}{2} \|L\|_1$$

$$\leq \frac{f(x_1) - f(x^*)}{T\eta\alpha_l} + \frac{\alpha_u \|\tilde{\sigma}\|_1}{\alpha_l \sqrt{b}} + \frac{\eta \alpha_u^2}{2\alpha_l} \|L\|_1.$$

$\square$

## C  COMPARISON OF CONVERGENCE RATES OF LARS AND SGD

Inspired by the comparison used by (Bernstein et al., 2018) for comparing SIGN SGD with SGD, we define the following quantities:

$$\left( \sum_{i=1}^h \|\nabla_i f(x_t)\| \right)^2 = \frac{\psi(\nabla f(x_t)) d \|\nabla f(x_t)\|^2}{h} \geq \frac{\psi_g d \|\nabla f(x_t)\|^2}{h}$$

$$\|L\|_1^2 \leq \frac{\psi_L d^2 \|L\|_\infty^2}{h^2}$$

$$\|\sigma\|_1^2 = \frac{\psi_\sigma d \|\sigma\|^2}{h}.$$

Then LARS convergence rate can be written in the following manner:

$$(\mathbb{E}[\|\nabla f(x_a)\|])^2 \leq O\left( \frac{(f(x_1) - f(x^*)) L_\infty}{T} \frac{\psi_L}{\psi_g^2} + \frac{\|\sigma\|^2}{T} \frac{\psi_\sigma^2}{\psi_g^2} \right).$$

If $\psi_L \ll \psi_g^2$ and $\psi_\sigma \ll \psi_g^2$ then LARS (i.e., gradient is more denser than curvature or stochasticity), we gain over SGD. Otherwise, SGD's upper bound on convergence rate is better.

## D  N-LAMB: NESTEROV MOMENTUM FOR LAMB

Sutskever et al. (2013) report that Nesterov's accelerated gradient (NAG) proposed by Nesterov (1983) is conceptually and empirically better than the regular momentum method for convex, non-stochastic objectives. Dozat (2016) incorporated Nesterov's momentum into Adam optimizer and proposed the Nadam optimizer. Specifically, only the first moment of Adam was modified and the second moment of Adam was unchanged. The results on several applications (Word2Vec, Image Recognition,

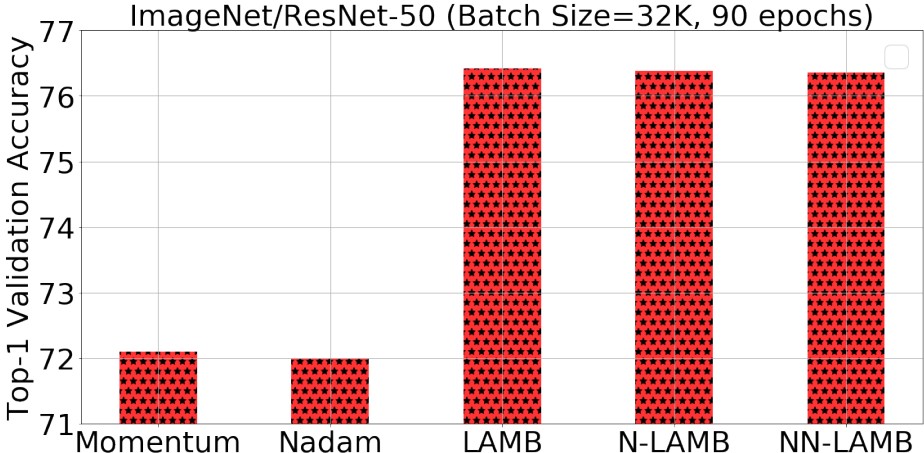

Figure 1: This figure shows N-LAMB and NN-LAMB can achieve a comparable accuracy compared to LAMB optimizer. Their performances are much better than momentum solver. The result of momentum optimizer was reported by Goyal et al. (2017). For Nadam, we use the learning rate recipe of (Goyal et al., 2017): (1) 5-epoch warmup to stablize the initial stage; and (2) multiply the learning rate by 0.1 at 30th, 60th, and 80th epoch. The target accuracy is around 0.763 (Goyal et al., 2017). We also tuned the learning rate of Nadam in {1e-4, 2e-4, ..., 9e-4, 1e-3, 2e-3, ..., 9e-3, 1e-2}.

and LSTM Language Model) showed that Nadam optimizer improves the speed of convergence and the quality of the learned models. We also tried using Nesterov's momentum to replace the regular momentum of LAMB optimizer's first moment. In this way, we got a new algorithm named as N-LAMB (Nesterov LAMB). The complete algorithm is in Algorithm 3. We can also Nesterov's momentum to replace the regular momentum of LAMB optimizer's second moment. We refer to this algorithm as NN-LAMB (Nesterov's momentum for both the first moment and the second moment). The details of NN-LAMB were shown in Algorithm 4.

Dozat (2016) suggested the best performance of Nadam was achieved by $\beta_1 = 0.975$, $\beta_2 = 0.999$, and $\epsilon = 1e-8$. We used the same settings for N-LAMB and NN-LAMB. We scaled the batch size to 32K for ImageNet training with ResNet-50. Our experimental results show that N-LAMB and NN-LAMB can achieve a comparable accuracy compared to LAMB optimizer. Their performances are much better than momentum solver (Figure 1).

## E   LAMB WITH LEARNING RATE CORRECTION

There are two operations at each iteration in original Adam optimizer (let us call it adam-correction):

$$m_t = m_t/(1 - \beta_1^t)$$

$$v_t = v_t/(1 - \beta_2^t)$$

It has an impact on the learning rate by $\eta_t := \eta_t * \sqrt{(1 - \beta_2^t)/(1 - \beta_1^t)}$. According to our experimental results, adam-correction essentially has the same effect as learning rate warmup (see Figure 2). The warmup function often was implemented in the modern deep learning system. Thus, we can remove adam-correction from the LAMB optimizer. We did not observe any drop in the test or validation accuracy for BERT and ImageNet training.

## F   LAMB WITH DIFFERENT NORMS

We need to compute the matrix/tensor norm for each layer when we do the parameter updating in the LAMB optimizer. We tried different norms in LAMB optimizer. However, we did not observe a significant difference in the validation accuracy of ImageNet training with ResNet-50. In our experiments, the difference in validation accuracy is less than 0.1 percent (Figure 3). We use L2 norm as the default.

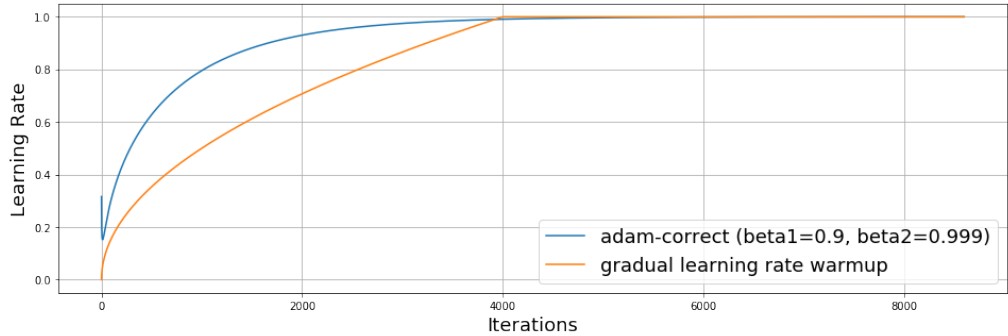

Figure 2: The figure shows that adam-correction has the same effect as learning rate warmup. We removed adam-correction from the LAMB optimizer. We did not observe any drop in the test or validation accuracy for BERT and ImageNet training.

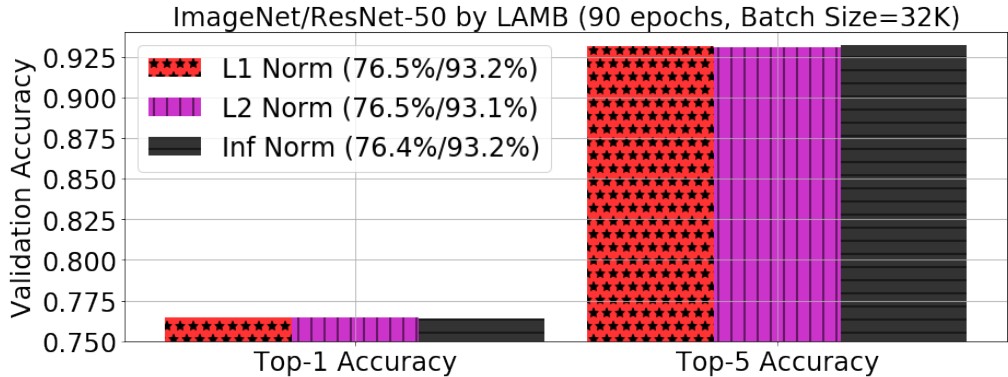

Figure 3: We tried different norms in LAMB optimizer. However, we did not observe a significant difference in the validation accuracy of ImageNet training with ResNet-50. We use L2 norm as the default.

## G    REGULAR BATCH SIZES FOR SMALL DATASETS: MNIST AND CIFAR-10.

According to DAWNBench, DavidNet (a custom 9-layer Residual ConvNet) is the fastest model for CIFAR-10 dataset (as of April 1st, 2019)[5]. The baseline uses the momentum SGD optimizer. Table 6 and Figure 4 show the test accuracy of CIFAR-10 training with DavidNet. The PyTorch implementation (momentum SGD optimizer) on GPUs was reported on Standford DAWNBench's website, which achieves 94.06% in 24 epochs. The Tensorflow implementation (momentum SGD optimizer) on TPU achieves a 93.72% accuracy in 24 epochs[6]. We use the implementation of TensorFlow on TPUs. LAMB optimizer is able to achieve 94.08% test accuracy in 24 epochs, which is better than other adaptive optimizers and momentum SGD. Even on the smaller tasks like MNIST training with LeNet, LAMB is able to achieve a better accuracy than existing solvers (Table 7).

## H    IMPLEMENTATION DETAILS AND ADDITIONAL RESULTS

There are several hyper-parameters in LAMB optimizer. Although users do not need to tune them, we explain them to help users to have a better understanding. $\beta_1$ is used for decaying the running average of the gradient. $\beta_2$ is used for decaying the running average of the square of gradient. The default setting for other parameters: weight decay rate $\lambda$=0.01, $\beta_1$=0.9, $\beta_2$=0.999, $\epsilon$=1e-6. We did not tune $\beta_1$ and $\beta_2$. However, our experiments show that tuning them may get a higher accuracy.

---

[5]https://dawn.cs.stanford.edu/benchmark/CIFAR10/train.html
[6]https://github.com/fenwickslab/dl_tutorials/blob/master/tutorial3_cifar10_davidnet_fix.ipynb

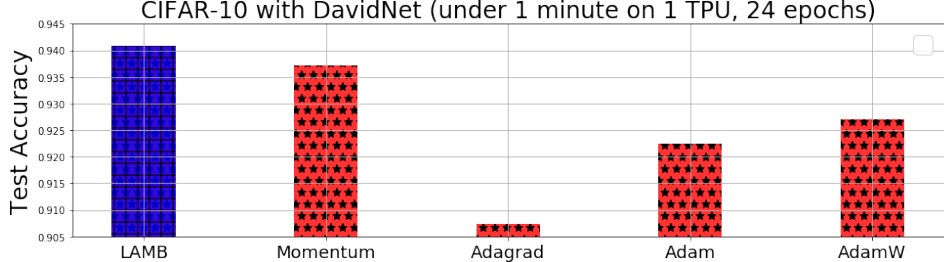

Figure 4: LAMB is better than the existing solvers (batch size = 512). We make sure all the solvers are carefully tuned. The learning rate tuning space of Adam, AdamW, Adagrad and LAMB is {0.0001, 0.0002, 0.0004, 0.0006, 0.0008, 0.001, 0.002, 0.004, 0.006, 0.008, 0.01, 0.02, 0.04, 0.06, 0.08, 0.1, 0.2, 0.4, 0.6, 0.8, 1, 2, 4, 6, 8, 10, 15, 20, 25, 30, 35, 40, 45, 50}. The momentum optimizer was tuned by the baseline implementer. The weight decay term of AdamW was tuned by {0.0001, 0.001, 0.01, 0.1, 1.0}.

Table 6: CIFAR-10 training with DavidNet (batch size = 512). All of them run 24 epochs and finish the training under one minute on one cloud TPU. We make sure all the solvers are carefully tuned. The learning rate tuning space of Adam, AdamW, Adagrad and LAMB is {0.0001, 0.0002, 0.0004, 0.0006, 0.0008, 0.001, 0.002, 0.004, 0.006, 0.008, 0.01, 0.02, 0.04, 0.06, 0.08, 0.1, 0.2, 0.4, 0.6, 0.8, 1, 2, 4, 6, 8, 10, 15, 20, 25, 30, 35, 40, 45, 50}. The momentum optimizer was tuned by the baseline implementer. The weight decay term of AdamW was tuned by {0.0001, 0.001, 0.01, 0.1, 1.0}.

| Optimizer | ADAGRAD | ADAM | ADAMW | momentum | LAMB |
|---|---|---|---|---|---|
| Test Accuracy | 0.9074 | 0.9225 | 0.9271 | 0.9372 | 0.9408 |

Based on our experience, learning rate is the most important hyper-parameter that affects the learning efficiency and final accuracy. Bengio (2012) suggests that it is often the single most important hyper-parameter and that it always should be tuned. Thus, to make sure we have a solid baseline, we carefully tune the learning rate of ADAM, ADAMW, ADAGRAD, and momentum SGD

In our experiments, we found that the validation loss is not reliable for large-batch training. A lower validation loss does not necessarily lead to a higher validation accuracy (Figure 5). Thus, we use the test/val accuracy or F1 score on dev set to evaluate the optimizers.

### H.0.1 BERT

Table 8 shows some of the tuning information from BERT training with ADAMW optimizer. ADAMW stops scaling at the batch size of 16K. The target F1 score is 90.5. LAMB achieves a F1 score of 91.345. The table shows the tuning information of ADAMW. In Table 8, we report the best F1 score we observed from our experiments.

The loss curves of BERT training by LAMB for different batch sizes are shown in Figure 6. We observe that the loss curves are almost identical to each other, which means our optimizer scales well with the batch size.

The training loss curve of BERT mixed-batch pre-training with LAMB is shown in Figure 7. This figure shows that LAMB can make the training converge smoothly at the batch size of 64K.

Figure 8 shows that we can achieve 76.8% scaling efficiency by scaling the batch size (49.1 times speedup by 64 times computational resources) and 101.8% scaling efficiency with mixed-batch (65.2 times speedup by 64 times computational resources)

### H.0.2 IMAGENET

Figures 9 - 14 show the LAMB trust ratio at different iterations for ImageNet training with ResNet-50. From these figures we can see that these ratios are very different from each other for different layers. LAMB uses the trust ratio to help the slow learners to train faster.

Table 7: Test Accuracy by MNIST training with LeNet (30 epochs for Batch Size = 1024). The tuning space of learning rate for all the optimizers is {0.0001, 0.001, 0.01, 0.1}. We use the same learning rate warmup and decay schedule for all of them.

| Optimizer | Momentum | Addgrad | ADAM | ADAMW | LAMB |
|---|---|---|---|---|---|
| Average accuracy over 5 runs | 0.9933 | 0.9928 | 0.9936 | 0.9941 | 0.9945 |

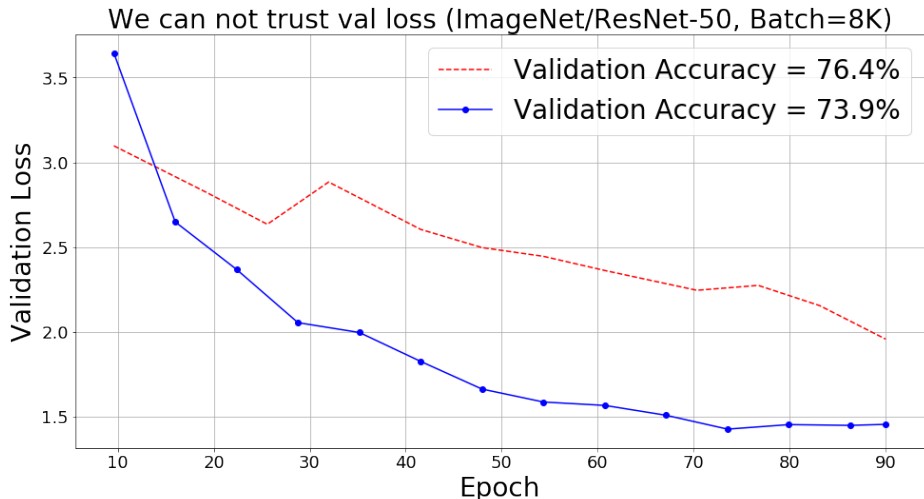

Figure 5: Our experiments show that even the validation loss is not reliable in the large-scale training. A lower validation loss may lead to a worse accuracy. Thus, we use the test/val accuracy or F1 score on dev set to evaluate the optimizers.

## H.1 BASELINE TUNING DETAILS FOR IMAGENET TRAINING WITH RESNET-50

If you are not interested in the baseline tuning details, please skip this section.

Goyal et al. (2017) suggested a proper learning rate warmup and decay scheme may help improve the ImageNet classification accuracy. We included these techniques in Adam/AdamW/AdaGrad tuning. Specifically, we use the learning rate recipe of Goyal et al. (2017): (1) 5-epoch warmup to stablize the initial stage; and (2) multiply the learning rate by 0.1 at 30th, 60th, and 80th epoch. The target accuracy is around 76.3% (Goyal et al., 2017). There techniques help to improve the accuracy of Adam/AdamW/AdaGrad to around 73%. However, even with these techniques, Adam/AdamW/AdaGrad stil can not achieve the target validation accuracy.

To make sure our baseline is solid, we carefully tuned the hyper-parameters. Table 9 shows the tuning information of standard Adagrad. Table 10 shows the tuning information of adding the learning rate scheme of Goyal et al. (2017) to standard Adagrad. Table 11 shows the tuning information of standard Adam. Table shows the tuning information of adding the learning rate scheme of Goyal et al. (2017) to standard Adam. It is tricky to tune the AdamW optimizer since both the L2 regularization and weight decay have the effect on the performance. Thus we have four tuning sets.

The first tuning set is based on AdamW with default L2 regularization. We tune the learning rate and weight decay. The tuning information is in Figures 13, 14, 15, and 16.

The second tuning set is based on AdamW with disabled L2 regularization. We tune the learning rate and weight decay. The tuning information is in Figures 17, 18, 19, and 20.

Then we add the learning rate scheme of Goyal et al. (2017) to AdamW and refer to it as AdamW+.

The third tuning set is based on AdamW+ with default L2 regularization. We tune the learning rate and weight decay. The tuning information is Figure 21 and 22.

The fourth tuning set is based on AdamW+ with disabled L2 regularization. We tune the learning rate and weight decay. The tuning information is in Figures 23, 24, 25.

Table 8: ADAMW stops scaling at the batch size of 16K. The target F1 score is 90.5. LAMB achieves a F1 score of 91.345. The table shows the tuning information of ADAMW. In this table, we report the best F1 score we observed from our experiments.

| Solver | batch size | warmup steps | LR | last step infomation | F1 score on dev set |
|---|---|---|---|---|---|
| ADAMW | 16K | 0.05×31250 | 0.0001 | loss=8.04471, step=28126 | diverged |
| ADAMW | 16K | 0.05×31250 | 0.0002 | loss=7.89673, step=28126 | diverged |
| ADAMW | 16K | 0.05×31250 | 0.0003 | loss=8.35102, step=28126 | diverged |
| ADAMW | 16K | 0.10×31250 | 0.0001 | loss=2.01419, step=31250 | 86.034 |
| ADAMW | 16K | 0.10×31250 | 0.0002 | loss=1.04689, step=31250 | 88.540 |
| ADAMW | 16K | 0.10×31250 | 0.0003 | loss=8.05845, step=20000 | diverged |
| ADAMW | 16K | 0.20×31250 | 0.0001 | loss=1.53706, step=31250 | 85.231 |
| ADAMW | 16K | 0.20×31250 | 0.0002 | loss=1.15500, step=31250 | 88.110 |
| ADAMW | 16K | 0.20×31250 | 0.0003 | loss=1.48798, step=31250 | 85.653 |

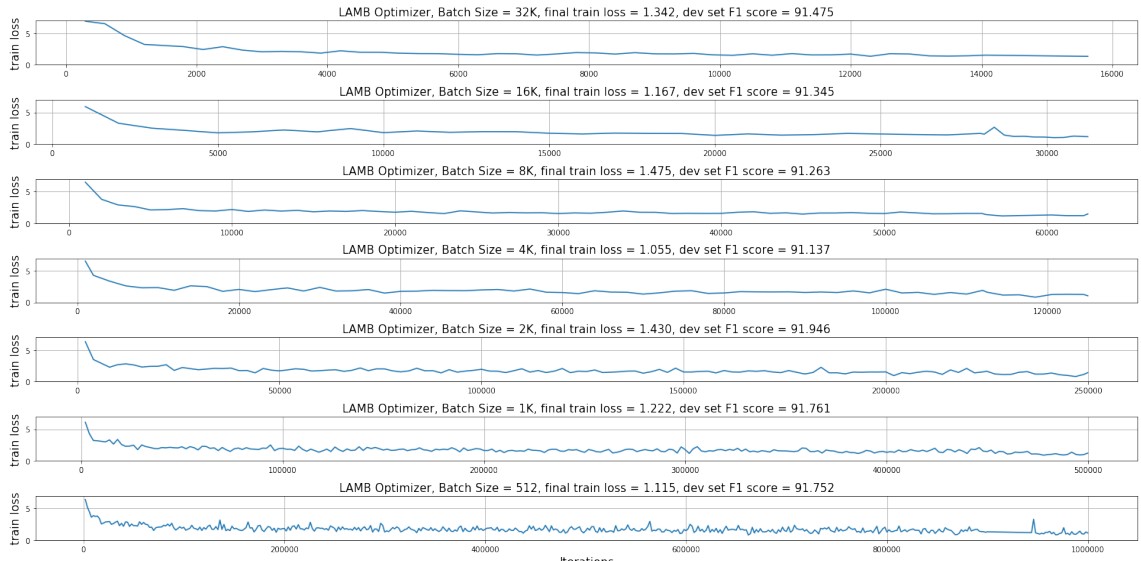

Figure 6: This figure shows the training loss curve of LAMB optimizer. We just want to use this figure to show that LAMB can make the training converge smoothly. Even if we scale the batch size to the extremely large cases, the loss curves are almost identical to each other.

Based on our comprehensive tuning results, we conclude the existing adaptive solvers do not perform well on ImageNet training or at least it is hard to tune them.

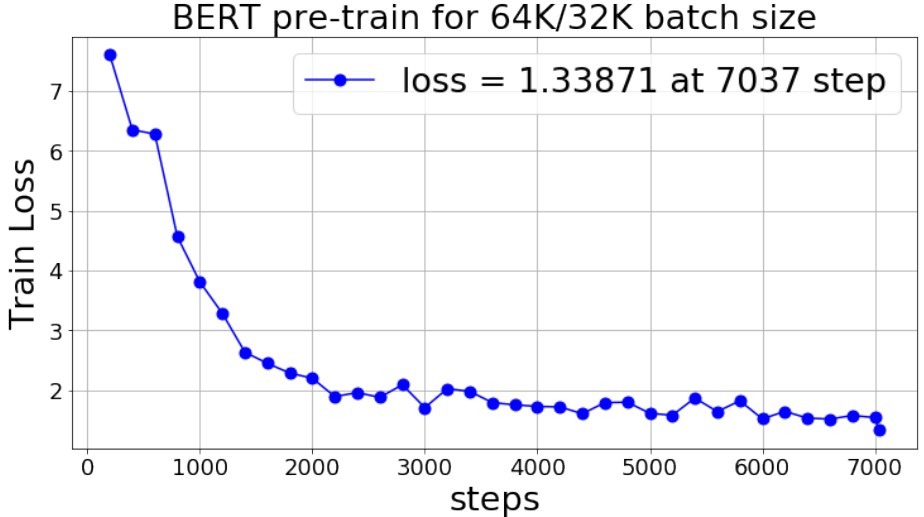

Figure 7: This figure shows the training loss curve of LAMB optimizer. This figure shows that LAMB can make the training converge smoothly at the extremely large batch size (e.g. 64K).

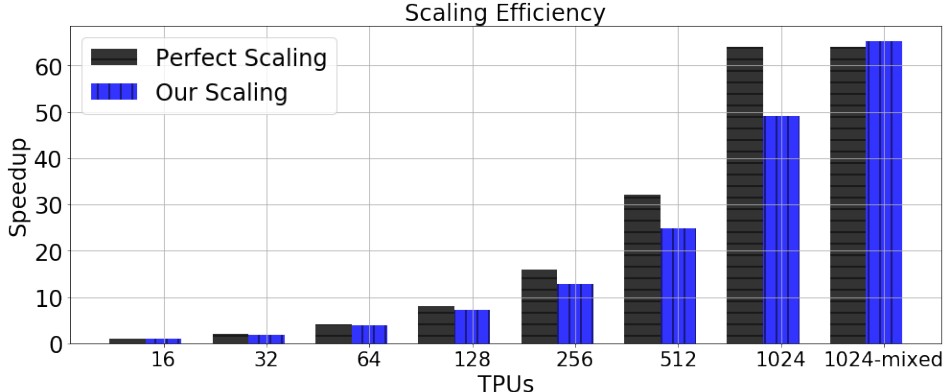

Figure 8: We achieve 76.8% scaling efficiency (49 times speedup by 64 times computational resources) and 101.8% scaling efficiency with a mixed, scaled batch size (65.2 times speedup by 64 times computational resources). 1024-mixed means the mixed-batch training on 1024 TPUs.

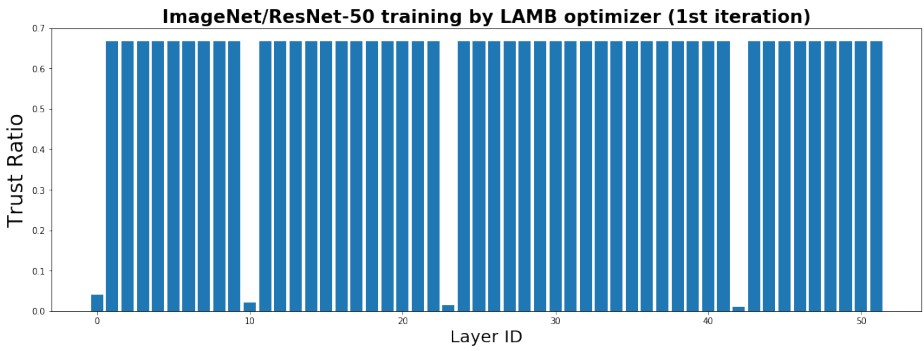

Figure 9: The LAMB trust ratio.

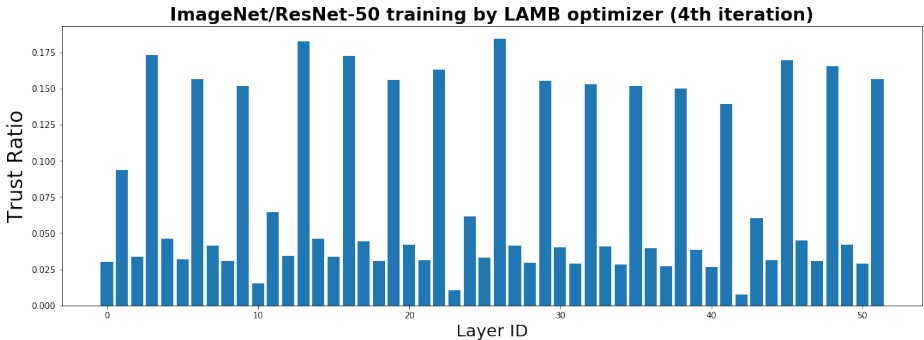

Figure 10: The LAMB trust ratio.

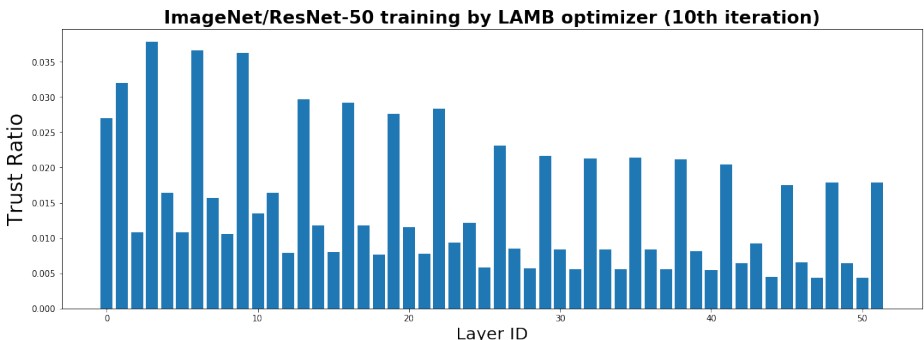

Figure 11: The LAMB trust ratio.

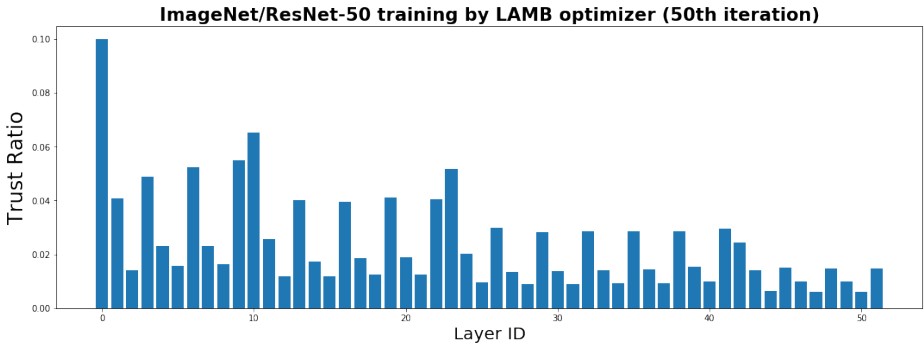

Figure 12: The LAMB trust ratio.

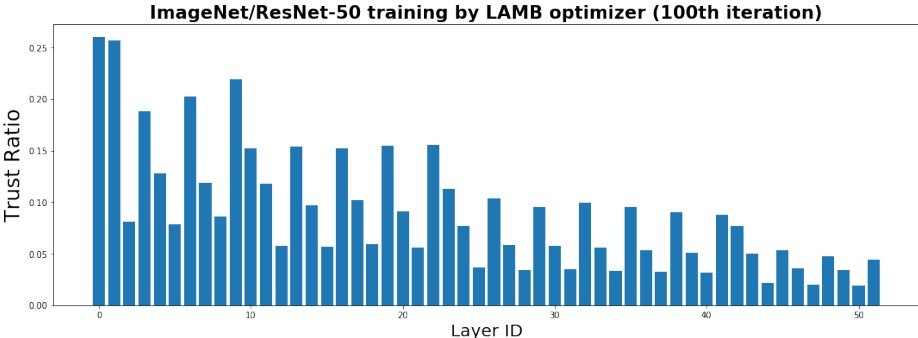

Figure 13: The LAMB trust ratio.

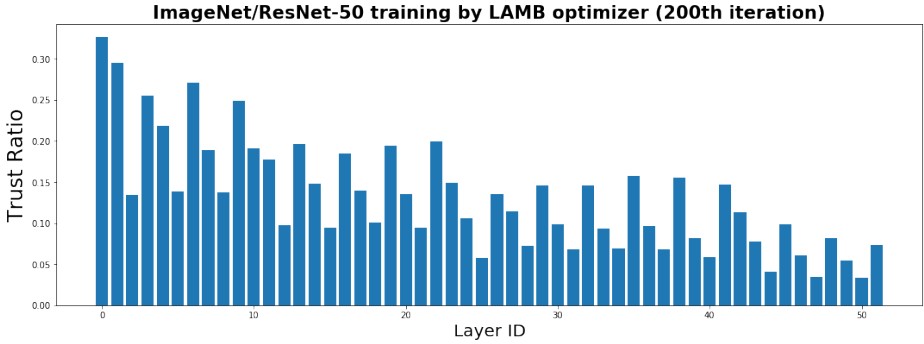

Figure 14: The LAMB trust ratio.

Table 9: The accuracy information of tuning default AdaGrad optimizer for ImageNet training with ResNet-50 (batch size = 16384, 90 epochs, 7038 iterations).

| Learning Rate | Top-1 Validation Accuracy |
|---|---|
| 0.0001 | 0.0026855469 |
| 0.001 | 0.015563965 |
| 0.002 | 0.022684732 |
| 0.004 | 0.030924479 |
| 0.008 | 0.04486084 |
| 0.010 | 0.054158527 |
| 0.020 | 0.0758667 |
| 0.040 | 0.1262614 |
| 0.080 | 0.24037679 |
| 0.100 | 0.27357993 |
| 0.200 | 0.458313 |
| 0.400 | **0.553833** |
| 0.800 | 0.54103595 |
| 1.000 | 0.5489095 |
| 2.000 | 0.47680664 |
| 4.000 | 0.5295207 |
| 6.000 | 0.36950684 |
| 8.000 | 0.31081137 |
| 10.00 | 0.30670166 |
| 12.00 | 0.3091024 |
| 14.00 | 0.3227946 |
| 16.00 | 0.0063680015 |
| 18.00 | 0.11287435 |
| 20.00 | 0.21602376 |
| 30.00 | 0.08315023 |
| 40.00 | 0.0132039385 |
| 50.00 | 0.0009969076 |

Table 10: The accuracy information of tuning AdaGrad optimizer for ImageNet training with ResNet-50 (batch size = 16384, 90 epochs, 7038 iterations). We use the learning rate recipe of (Goyal et al., 2017): (1) 5-epoch warmup to stablize the initial stage; and (2) multiply the learning rate by 0.1 at 30th, 60th, and 80th epoch. The target accuracy is around 0.763 (Goyal et al., 2017).

| Learning Rate | Top-1 Validation Accuracy |
| --- | --- |
| 0.0001 | 0.0011189779 |
| 0.001 | 0.00793457 |
| 0.002 | 0.012573242 |
| 0.004 | 0.019022623 |
| 0.008 | 0.027079264 |
| 0.010 | 0.029012045 |
| 0.020 | 0.0421346 |
| 0.040 | 0.06618246 |
| 0.080 | 0.10970052 |
| 0.100 | 0.13429768 |
| 0.200 | 0.26550293 |
| 0.400 | 0.41918945 |
| 0.800 | 0.5519816 |
| 1.000 | 0.58614093 |
| 2.000 | 0.67252606 |
| 4.000 | 0.70306396 |
| 6.000 | 0.709493 |
| 8.000 | 0.7137858 |
| 10.00 | 0.71797687 |
| 12.00 | 0.7187703 |
| 14.00 | **0.72007245** |
| 16.00 | 0.7194214 |
| 18.00 | 0.7149251 |
| 20.00 | 0.71293133 |
| 30.00 | 0.70458984 |
| 40.00 | 0.69085693 |
| 50.00 | 0.67976886 |

Table 11: The accuracy information of tuning default Adam optimizer for ImageNet training with ResNet-50 (batch size = 16384, 90 epochs, 7038 iterations). The target accuracy is around 0.763 (Goyal et al., 2017).

| Learning Rate | Top-1 Validation Accuracy |
|---|---|
| 0.0001 | 0.5521 |
| 0.0002 | 0.6089 |
| 0.0004 | 0.6432 |
| 0.0006 | 0.6465 |
| 0.0008 | 0.6479 |
| 0.001 | **0.6604** |
| 0.002 | 0.6408 |
| 0.004 | 0.5687 |
| 0.006 | 0.5165 |
| 0.008 | 0.4812 |
| 0.010 | 0.3673 |

Table 12: The accuracy information of tuning Adam optimizer for ImageNet training with ResNet-50 (batch size = 16384, 90 epochs, 7038 iterations). We use the learning rate recipe of (Goyal et al., 2017): (1) 5-epoch warmup to stablize the initial stage; and (2) multiply the learning rate by 0.1 at 30th, 60th, and 80th epoch. The target accuracy is around 0.763 (Goyal et al., 2017).

| Learning Rate | Top-1 Validation Accuracy |
|---|---|
| 0.0001 | 0.410319 |
| 0.0002 | 0.55263263 |
| 0.0004 | 0.6455485 |
| 0.0006 | 0.6774495 |
| 0.0008 | 0.6996867 |
| 0.001 | 0.71010333 |
| 0.002 | **0.73476154** |
| 0.004 | 0.73286945 |
| 0.006 | 0.72648114 |
| 0.008 | 0.72214764 |
| 0.010 | 0.71466064 |
| 0.012 | 0.7081502 |
| 0.014 | 0.6993001 |
| 0.016 | 0.69108075 |
| 0.020 | 0.67997235 |
| 0.040 | 0.58658856 |
| 0.060 | 0.51090497 |
| 0.080 | 0.45174155 |
| 0.100 | 0.40297446 |

Table 13: The accuracy information of tuning default AdamW optimizer for ImageNet training with ResNet-50 (batch size = 16384, 90 epochs, 7038 iterations). The target accuracy is around 0.763 (Goyal et al., 2017).

| learning rate | weight decay | L2 regularization | Top-1 Validation Accuracy |
|---|---|---|---|
| 0.0001 | 0.00001 | default (0.01) | 0.53312176 |
| 0.0002 | 0.00001 | default (0.01) | 0.5542806 |
| 0.0004 | 0.00001 | default (0.01) | 0.48769125 |
| 0.0006 | 0.00001 | default (0.01) | 0.46317545 |
| 0.0008 | 0.00001 | default (0.01) | 0.40903726 |
| 0.001 | 0.00001 | default (0.01) | 0.42401123 |
| 0.002 | 0.00001 | default (0.01) | 0.33870444 |
| 0.004 | 0.00001 | default (0.01) | 0.12339274 |
| 0.006 | 0.00001 | default (0.01) | 0.122924805 |
| 0.008 | 0.00001 | default (0.01) | 0.08099365 |
| 0.010 | 0.00001 | default (0.01) | 0.016764322 |
| 0.012 | 0.00001 | default (0.01) | 0.032714844 |
| 0.014 | 0.00001 | default (0.01) | 0.018147787 |
| 0.016 | 0.00001 | default (0.01) | 0.0066731772 |
| 0.018 | 0.00001 | default (0.01) | 0.010294597 |
| 0.020 | 0.00001 | default (0.01) | 0.008260091 |
| 0.025 | 0.00001 | default (0.01) | 0.008870442 |
| 0.030 | 0.00001 | default (0.01) | 0.0064493814 |
| 0.040 | 0.00001 | default (0.01) | 0.0018107096 |
| 0.050 | 0.00001 | default (0.01) | 0.003540039 |

Table 14: The accuracy information of tuning default AdamW optimizer for ImageNet training with ResNet-50 (batch size = 16384, 90 epochs, 7038 iterations). The target accuracy is around 0.763 (Goyal et al., 2017).

| learning rate | weight decay | L2 regularization | Top-1 Validation Accuracy |
|---|---|---|---|
| 0.0001 | 0.0001 | default (0.01) | 0.55489093 |
| 0.0002 | 0.0001 | default (0.01) | **0.56514484** |
| 0.0004 | 0.0001 | default (0.01) | 0.4986979 |
| 0.0006 | 0.0001 | default (0.01) | 0.47595215 |
| 0.0008 | 0.0001 | default (0.01) | 0.44685873 |
| 0.001 | 0.0001 | default (0.01) | 0.41029868 |
| 0.002 | 0.0001 | default (0.01) | 0.2808024 |
| 0.004 | 0.0001 | default (0.01) | 0.08111572 |
| 0.006 | 0.0001 | default (0.01) | 0.068115234 |
| 0.008 | 0.0001 | default (0.01) | 0.057922363 |
| 0.010 | 0.0001 | default (0.01) | 0.05222575 |
| 0.012 | 0.0001 | default (0.01) | 0.017313639 |
| 0.014 | 0.0001 | default (0.01) | 0.029785156 |
| 0.016 | 0.0001 | default (0.01) | 0.016540527 |
| 0.018 | 0.0001 | default (0.01) | 0.00575765 |
| 0.020 | 0.0001 | default (0.01) | 0.0102335615 |
| 0.025 | 0.0001 | default (0.01) | 0.0060831704 |
| 0.030 | 0.0001 | default (0.01) | 0.0036417644 |
| 0.040 | 0.0001 | default (0.01) | 0.0010782877 |
| 0.050 | 0.0001 | default (0.01) | 0.0037638347 |

Table 15: The accuracy information of tuning default AdamW optimizer for ImageNet training with ResNet-50 (batch size = 16384, 90 epochs, 7038 iterations). The target accuracy is around 0.763 (Goyal et al., 2017).

| learning rate | weight decay | L2 regularization | Top-1 Validation Accuracy |
|---|---|---|---|
| 0.0001 | 0.001 | default (0.01) | 0.21142578 |
| 0.0002 | 0.001 | default (0.01) | 0.4289144 |
| 0.0004 | 0.001 | default (0.01) | 0.13537598 |
| 0.0006 | 0.001 | default (0.01) | 0.33803305 |
| 0.0008 | 0.001 | default (0.01) | 0.32611084 |
| 0.001 | 0.001 | default (0.01) | 0.22194417 |
| 0.002 | 0.001 | default (0.01) | 0.1833903 |
| 0.004 | 0.001 | default (0.01) | 0.08256022 |
| 0.006 | 0.001 | default (0.01) | 0.020507812 |
| 0.008 | 0.001 | default (0.01) | 0.018269857 |
| 0.010 | 0.001 | default (0.01) | 0.007507324 |
| 0.012 | 0.001 | default (0.01) | 0.020080566 |
| 0.014 | 0.001 | default (0.01) | 0.010762532 |
| 0.016 | 0.001 | default (0.01) | 0.0021362305 |
| 0.018 | 0.001 | default (0.01) | 0.007954915 |
| 0.020 | 0.001 | default (0.01) | 0.005859375 |
| 0.025 | 0.001 | default (0.01) | 0.009724935 |
| 0.030 | 0.001 | default (0.01) | 0.0019124349 |
| 0.040 | 0.001 | default (0.01) | 0.00390625 |
| 0.050 | 0.001 | default (0.01) | 0.0009969076 |

Table 16: The accuracy information of tuning default AdamW optimizer for ImageNet training with ResNet-50 (batch size = 16384, 90 epochs, 7038 iterations). The target accuracy is around 0.763 (Goyal et al., 2017).

| learning rate | weight decay | L2 regularization | Top-1 Validation Accuracy |
|---------------|--------------|-------------------|---------------------------|
| 0.0001 | 0.01 | default (0.01) | 0.0009765625 |
| 0.0002 | 0.01 | default (0.01) | 0.0009969076 |
| 0.0004 | 0.01 | default (0.01) | 0.0010172526 |
| 0.0006 | 0.01 | default (0.01) | 0.0009358724 |
| 0.0008 | 0.01 | default (0.01) | 0.0022379558 |
| 0.001 | 0.01 | default (0.01) | 0.001566569 |
| 0.002 | 0.01 | default (0.01) | 0.009480794 |
| 0.004 | 0.01 | default (0.01) | 0.0033569336 |
| 0.006 | 0.01 | default (0.01) | 0.0029907227 |
| 0.008 | 0.01 | default (0.01) | 0.0018513998 |
| 0.010 | 0.01 | default (0.01) | 0.009134929 |
| 0.012 | 0.01 | default (0.01) | 0.0022176106 |
| 0.014 | 0.01 | default (0.01) | 0.0040690103 |
| 0.016 | 0.01 | default (0.01) | 0.0017293295 |
| 0.018 | 0.01 | default (0.01) | 0.00061035156 |
| 0.020 | 0.01 | default (0.01) | 0.0022379558 |
| 0.025 | 0.01 | default (0.01) | 0.0017089844 |
| 0.030 | 0.01 | default (0.01) | 0.0014241537 |
| 0.040 | 0.01 | default (0.01) | 0.0020345051 |
| 0.050 | 0.01 | default (0.01) | 0.0012817383 |

Table 17: The accuracy information of tuning default AdamW optimizer for ImageNet training with ResNet-50 (batch size = 16384, 90 epochs, 7038 iterations). The target accuracy is around 0.763 (Goyal et al., 2017).

| learning rate | weight decay | L2 regularization | Top-1 Validation Accuracy |
|---------------|--------------|-------------------|---------------------------|
| 0.0001 | 0.00001 | disable | 0.48917642 |
| 0.0002 | 0.00001 | disable | 0.58152264 |
| 0.0004 | 0.00001 | disable | 0.63460284 |
| 0.0006 | 0.00001 | disable | 0.64849854 |
| 0.0008 | 0.00001 | disable | 0.6598918 |
| 0.001 | 0.00001 | disable | 0.6662801 |
| 0.002 | 0.00001 | disable | **0.67266846** |
| 0.004 | 0.00001 | disable | 0.6692708 |
| 0.006 | 0.00001 | disable | 0.6573079 |
| 0.008 | 0.00001 | disable | 0.6639404 |
| 0.010 | 0.00001 | disable | 0.65230304 |
| 0.012 | 0.00001 | disable | 0.6505534 |
| 0.014 | 0.00001 | disable | 0.64990234 |
| 0.016 | 0.00001 | disable | 0.65323895 |
| 0.018 | 0.00001 | disable | 0.67026776 |
| 0.020 | 0.00001 | disable | 0.66086835 |
| 0.025 | 0.00001 | disable | 0.65425617 |
| 0.030 | 0.00001 | disable | 0.6476237 |
| 0.040 | 0.00001 | disable | 0.55478925 |
| 0.050 | 0.00001 | disable | 0.61869305 |

Table 18: The accuracy information of tuning default AdamW optimizer for ImageNet training with ResNet-50 (batch size = 16384, 90 epochs, 7038 iterations). The target accuracy is around 0.763 (Goyal et al., 2017).

| learning rate | weight decay | L2 regularization | Top-1 Validation Accuracy |
|---|---|---|---|
| 0.0001 | 0.0001 | disable | 0.5033366 |
| 0.0002 | 0.0001 | disable | 0.5949707 |
| 0.0004 | 0.0001 | disable | 0.62561035 |
| 0.0006 | 0.0001 | disable | 0.6545207 |
| 0.0008 | 0.0001 | disable | 0.66326904 |
| 0.001 | 0.0001 | disable | 0.6677043 |
| 0.002 | 0.0001 | disable | **0.67244464** |
| 0.004 | 0.0001 | disable | 0.6702881 |
| 0.006 | 0.0001 | disable | 0.66033936 |
| 0.008 | 0.0001 | disable | 0.66426593 |
| 0.010 | 0.0001 | disable | 0.66151935 |
| 0.012 | 0.0001 | disable | 0.6545817 |
| 0.014 | 0.0001 | disable | 0.65509033 |
| 0.016 | 0.0001 | disable | 0.6529338 |
| 0.018 | 0.0001 | disable | 0.65651447 |
| 0.020 | 0.0001 | disable | 0.65334064 |
| 0.025 | 0.0001 | disable | 0.655009 |
| 0.030 | 0.0001 | disable | 0.64552814 |
| 0.040 | 0.0001 | disable | 0.6425374 |
| 0.050 | 0.0001 | disable | 0.5988159 |

Table 19: The accuracy information of tuning default AdamW optimizer for ImageNet training with ResNet-50 (batch size = 16384, 90 epochs, 7038 iterations). The target accuracy is around 0.763 (Goyal et al., 2017).

| learning rate | weight decay | L2 regularization | Top-1 Validation Accuracy |
|---------------|--------------|-------------------|---------------------------|
| 0.0001 | 0.001 | disable | 0.4611206 |
| 0.0002 | 0.001 | disable | 0.0076293945 |
| 0.0004 | 0.001 | disable | 0.29233804 |
| 0.0006 | 0.001 | disable | 0.57295734 |
| 0.0008 | 0.001 | disable | 0.5574748 |
| 0.001 | 0.001 | disable | 0.5988566 |
| 0.002 | 0.001 | disable | 0.586263 |
| 0.004 | 0.001 | disable | **0.62076825** |
| 0.006 | 0.001 | disable | 0.61503094 |
| 0.008 | 0.001 | disable | 0.4697876 |
| 0.010 | 0.001 | disable | 0.619751 |
| 0.012 | 0.001 | disable | 0.54243976 |
| 0.014 | 0.001 | disable | 0.5429077 |
| 0.016 | 0.001 | disable | 0.55281574 |
| 0.018 | 0.001 | disable | 0.5819295 |
| 0.020 | 0.001 | disable | 0.5938924 |
| 0.025 | 0.001 | disable | 0.541097 |
| 0.030 | 0.001 | disable | 0.45890298 |
| 0.040 | 0.001 | disable | 0.56193036 |
| 0.050 | 0.001 | disable | 0.5279134 |

Table 20: The accuracy information of tuning default AdamW optimizer for ImageNet training with ResNet-50 (batch size = 16384, 90 epochs, 7038 iterations). The target accuracy is around 0.763 (Goyal et al., 2017).

| learning rate | weight decay | L2 regularization | Top-1 Validation Accuracy |
|---|---|---|---|
| 0.0001 | 0.01 | disable | 0.0009969076 |
| 0.0002 | 0.01 | disable | 0.0008951823 |
| 0.0004 | 0.01 | disable | 0.00095621747 |
| 0.0006 | 0.01 | disable | 0.0012817383 |
| 0.0008 | 0.01 | disable | 0.016886393 |
| 0.001 | 0.01 | disable | 0.038146973 |
| 0.002 | 0.01 | disable | 0.0015258789 |
| 0.004 | 0.01 | disable | 0.0014241537 |
| 0.006 | 0.01 | disable | 0.081441246 |
| 0.008 | 0.01 | disable | 0.028116861 |
| 0.010 | 0.01 | disable | 0.011820476 |
| 0.012 | 0.01 | disable | 0.08138021 |
| 0.014 | 0.01 | disable | 0.010111491 |
| 0.016 | 0.01 | disable | 0.0041910806 |
| 0.018 | 0.01 | disable | 0.0038248699 |
| 0.020 | 0.01 | disable | 0.002746582 |
| 0.025 | 0.01 | disable | 0.011555989 |
| 0.030 | 0.01 | disable | 0.0065104165 |
| 0.040 | 0.01 | disable | 0.016438803 |
| 0.050 | 0.01 | disable | 0.007710775 |

Table 21: The accuracy information of tuning AdamW optimizer for ImageNet training with ResNet-50 (batch size = 16384, 90 epochs, 7038 iterations). We use the learning rate recipe of (Goyal et al., 2017): (1) 5-epoch warmup to stablize the initial stage; and (2) multiply the learning rate by 0.1 at 30th, 60th, and 80th epoch. The target accuracy is around 0.763 (Goyal et al., 2017).

| learning rate | weight decay | L2 regularization | Top-1 Validation Accuracy |
| --- | --- | --- | --- |
| 0.0001 | 0.01 | default (0.01) | 0.0009969076 |
| 0.0002 | 0.01 | default (0.01) | 0.0009969076 |
| 0.0004 | 0.01 | default (0.01) | 0.0009969076 |
| 0.0006 | 0.01 | default (0.01) | 0.0009358724 |
| 0.0008 | 0.01 | default (0.01) | 0.0009969076 |
| 0.001 | 0.01 | default (0.01) | 0.0009765625 |
| 0.002 | 0.01 | default (0.01) | 0.0010172526 |
| 0.004 | 0.01 | default (0.01) | 0.0010172526 |
| 0.006 | 0.01 | default (0.01) | 0.0010172526 |
| 0.008 | 0.01 | default (0.01) | 0.0010172526 |
| 0.0001 | 0.001 | default (0.01) | 0.0010172526 |
| 0.0002 | 0.001 | default (0.01) | 0.0010172526 |
| 0.0004 | 0.001 | default (0.01) | 0.0010172526 |
| 0.0006 | 0.001 | default (0.01) | 0.0009969076 |
| 0.0008 | 0.001 | default (0.01) | 0.0010172526 |
| 0.001 | 0.001 | default (0.01) | 0.0010172526 |
| 0.002 | 0.001 | default (0.01) | 0.0010172526 |
| 0.004 | 0.001 | default (0.01) | 0.0038452148 |
| 0.006 | 0.001 | default (0.01) | 0.011881511 |
| 0.008 | 0.001 | default (0.01) | 0.0061442056 |

Table 22: The accuracy information of tuning AdamW optimizer for ImageNet training with ResNet-50 (batch size = 16384, 90 epochs, 7038 iterations). We use the learning rate recipe of (Goyal et al., 2017): (1) 5-epoch warmup to stablize the initial stage; and (2) multiply the learning rate by 0.1 at 30th, 60th, and 80th epoch. The target accuracy is around 0.763 (Goyal et al., 2017).

| learning rate | weight decay | L2 regularization | Top-1 Validation Accuracy |
|---|---|---|---|
| 0.0001 | 0.0001 | default (0.01) | 0.3665975 |
| 0.0002 | 0.0001 | default (0.01) | 0.5315755 |
| 0.0004 | 0.0001 | default (0.01) | 0.6369222 |
| 0.0006 | 0.0001 | default (0.01) | 0.6760457 |
| 0.0008 | 0.0001 | default (0.01) | 0.69557697 |
| 0.001 | 0.0001 | default (0.01) | 0.7076009 |
| 0.002 | 0.0001 | default (0.01) | **0.73065186** |
| 0.004 | 0.0001 | default (0.01) | 0.72806805 |
| 0.006 | 0.0001 | default (0.01) | 0.72161865 |
| 0.008 | 0.0001 | default (0.01) | 0.71816 |
| 0.0001 | 0.00001 | default (0.01) | 0.49804688 |
| 0.0002 | 0.00001 | default (0.01) | 0.6287028 |
| 0.0004 | 0.00001 | default (0.01) | 0.6773885 |
| 0.0006 | 0.00001 | default (0.01) | 0.67348224 |
| 0.0008 | 0.00001 | default (0.01) | 0.6622111 |
| 0.001 | 0.00001 | default (0.01) | 0.6468709 |
| 0.002 | 0.00001 | default (0.01) | 0.5846761 |
| 0.004 | 0.00001 | default (0.01) | 0.4868978 |
| 0.006 | 0.00001 | default (0.01) | 0.34969077 |
| 0.008 | 0.00001 | default (0.01) | 0.31193033 |

Table 23: The accuracy information of tuning AdamW optimizer for ImageNet training with ResNet-50 (batch size = 16384, 90 epochs, 7038 iterations). We use the learning rate recipe of (Goyal et al., 2017): (1) 5-epoch warmup to stablize the initial stage; and (2) multiply the learning rate by 0.1 at 30th, 60th, and 80th epoch. The target accuracy is around 0.763 (Goyal et al., 2017).

| learning rate | weight decay | L2 regularization | Top-1 Validation Accuracy |
|---|---|---|---|
| 0.0001 | 0.01 | disable | 0.0010172526 |
| 0.0002 | 0.01 | disable | 0.0009765625 |
| 0.0004 | 0.01 | disable | 0.0010172526 |
| 0.0006 | 0.01 | disable | 0.0009969076 |
| 0.0008 | 0.01 | disable | 0.0010172526 |
| 0.001 | 0.01 | disable | 0.0009765625 |
| 0.002 | 0.01 | disable | 0.0009969076 |
| 0.004 | 0.01 | disable | 0.0009969076 |
| 0.006 | 0.01 | disable | 0.0009765625 |
| 0.008 | 0.01 | disable | 0.0010172526 |
| 0.0001 | 0.001 | disable | 0.0009765625 |
| 0.0002 | 0.001 | disable | 0.0010172526 |
| 0.0004 | 0.001 | disable | 0.0010172526 |
| 0.0006 | 0.001 | disable | 0.0010172526 |
| 0.0008 | 0.001 | disable | 0.0010172526 |
| 0.001 | 0.001 | disable | 0.0009969076 |
| 0.002 | 0.001 | disable | 0.0010579427 |
| 0.004 | 0.001 | disable | 0.0016886393 |
| 0.006 | 0.001 | disable | 0.019714355 |
| 0.008 | 0.001 | disable | 0.1329956 |

Table 24: The accuracy information of tuning AdamW optimizer for ImageNet training with ResNet-50 (batch size = 16384, 90 epochs, 7038 iterations). We use the learning rate recipe of (Goyal et al., 2017): (1) 5-epoch warmup to stablize the initial stage; and (2) multiply the learning rate by 0.1 at 30th, 60th, and 80th epoch. The target accuracy is around 0.763 (Goyal et al., 2017).

| learning rate | weight decay | L2 regularization | Top-1 Validation Accuracy |
|---|---|---|---|
| 0.0001 | 0.0001 | disable | 0.28515625 |
| 0.0002 | 0.0001 | disable | 0.44055176 |
| 0.0004 | 0.0001 | disable | 0.56815594 |
| 0.0006 | 0.0001 | disable | 0.6234741 |
| 0.0008 | 0.0001 | disable | 0.6530762 |
| 0.001 | 0.0001 | disable | 0.6695964 |
| 0.002 | 0.0001 | disable | 0.70048016 |
| 0.004 | 0.0001 | disable | 0.71698 |
| 0.006 | 0.0001 | disable | 0.72021484 |
| 0.008 | 0.0001 | disable | **0.7223918** |
| 0.010 | 0.0001 | disable | 0.72017413 |
| 0.012 | 0.0001 | disable | 0.72058105 |
| 0.014 | 0.0001 | disable | 0.7188924 |
| 0.016 | 0.0001 | disable | 0.71695966 |
| 0.018 | 0.0001 | disable | 0.7154134 |
| 0.020 | 0.0001 | disable | 0.71358234 |
| 0.025 | 0.0001 | disable | 0.7145386 |
| 0.030 | 0.0001 | disable | 0.7114258 |
| 0.040 | 0.0001 | disable | 0.7066447 |
| 0.050 | 0.0001 | disable | 0.70284015 |

Table 25: The accuracy information of tuning AdamW optimizer for ImageNet training with ResNet-50 (batch size = 16384, 90 epochs, 7038 iterations). We use the learning rate recipe of (Goyal et al., 2017): (1) 5-epoch warmup to stablize the initial stage; and (2) multiply the learning rate by 0.1 at 30th, 60th, and 80th epoch. The target accuracy is around 0.763 (Goyal et al., 2017).

| learning rate | weight decay | L2 regularization | Top-1 Validation Accuracy |
|---|---|---|---|
| 0.0001 | 0.00001 | disable | 0.31247965 |
| 0.0002 | 0.00001 | disable | 0.4534912 |
| 0.0004 | 0.00001 | disable | 0.57765704 |
| 0.0006 | 0.00001 | disable | 0.6277669 |
| 0.0008 | 0.00001 | disable | 0.65321857 |
| 0.001 | 0.00001 | disable | 0.6682129 |
| 0.002 | 0.00001 | disable | 0.69938153 |
| 0.004 | 0.00001 | disable | 0.7095947 |
| 0.006 | 0.00001 | disable | 0.710612 |
| 0.008 | 0.00001 | disable | 0.70857745 |
| 0.010 | 0.00001 | disable | 0.7094116 |
| 0.012 | 0.00001 | disable | 0.70717365 |
| 0.014 | 0.00001 | disable | **0.7109375** |
| 0.016 | 0.00001 | disable | 0.7058309 |
| 0.018 | 0.00001 | disable | 0.7052409 |
| 0.020 | 0.00001 | disable | 0.7064412 |
| 0.025 | 0.00001 | disable | 0.7035319 |
| 0.030 | 0.00001 | disable | 0.6994629 |
| 0.040 | 0.00001 | disable | 0.6972656 |
| 0.050 | 0.00001 | disable | 0.6971232 |

