# OpenReview forum: "Large Batch Optimization for Deep Learning: Training BERT in 76 minutes"
_ICLR.cc/2020/Conference — Accept (Poster)_

### Official Review · AnonReviewer1 · 2019-10-21
**Official Blind Review #1**

**Rating:** 3

**Review:**


In this paper, the authors made a study on large-batch training for the BERT, and successfully trained a BERT model in 76 minutes. The results look quite exciting, however, after looking into the details of the paper, I would say that this is just a kind of RED AI – the results were mostly achieved by putting together a huge number of TPUs, without necessary technical innovation and fundamental contributions.

1)	The work used 1024 TPUs to achieve 76-min training. If we compare this with the original BERT training (16 TPU for 81 hours), there is no algorithmic speed up at all (only system speedup). Not to mention that making a single BERT training faster by using more resources does not seem to be a big thing – one can do multiple BERT training experiments in parallel or in pipeline, which will correspond to similar innovation speed.

2)	The theoretical analysis is not very impressive, and to certain degree, is not helpful. The theory just says that in certain conditions, both LARS and LAMB converge fasters than SGD. However, LAMB ha no advantage over LARS at all, which cannot well explain the experimental observations. Furthermore, when \beta_2 > 0, the convergence rate of LAMB is even slower than LARS, which delivers some contradictory message. As we know, \beta_2>0 is very important, otherwise the optimization algorithm will not be ADAM at all.

Overall speaking, I am afraid that such work do not have sufficient theoretical or algorithmic contributions. And I doubt the true value of adding a huge number of computational resources to achieve speedup.


**I read the author rebuttal. Thanks for the clarification on the algorithmic contribution of LAMB. However, my other concerns still remain. I have adjusted my rating by a little, but I can hardly move to the positive side.

**Experience Assessment:**

I have published in this field for several years.

**Review Assessment: Checking Correctness Of Derivations And Theory:**

N/A

**Review Assessment: Checking Correctness Of Experiments:**

I carefully checked the experiments.

**Review Assessment: Thoroughness In Paper Reading:**

I read the paper thoroughly.

---

> ### Author Response · Authors · 2019-11-09
> **Response to Review #1**
>
> Thanks for your comments. We believe there is some confusion regarding the large batch setting in our work, which we clarify in detail below.
>
> Regarding the quote from first paragraph and comment 1: “The results were mostly achieved by putting together a huge number of TPUs, without necessary technical innovation and fundamental contributions.” “There’s no algorithm speed up (only system speedup).” “One can do multiple BERT training experiments in parallel or in pipeline, which will correspond to similar innovation speed.”
>
> We want to re-emphasize that the central goal of our paper is to develop an optimization technique that enables very large batch training for achieving the largest possible speedup (minimize the end-to-end wall time). Large batch training is a very challenging problem (please see references e.g., Keskar et al., 2016, Goyal et al., 2017) since naive approach of large batch optimization typically leads to poor generalization even after extensive hyperparameter tuning. Thus, simply increasing the batch size with more computational resources does not translate into good performance. This can also be seen from our experiments where simply increasing the batch size for Adam or LARS does not yield good performance (see discussion in Section 4.1 and Table 2).
>
> However, an appropriately designed optimization technique can make better use of larger batch size and reduce end-to-end wall time without hurting the accuracy. By using carefully selected per-parameter and layerwise learning rates, the proposed LAMB technique scales training to very large batch sizes without accuracy loss, and requires less tuning over differing batch sizes, loss functions, etc, which makes scaling up training much easier. With these novel normalization strategies, LAMB has enabled training many SOTA models in a scalable manner (which were previously hard to train). It is not clear if providing the relevant works will lead to some violation of double blind policy. But we will be happy to provide the references to relevant works if this is not the case.
>
> Regarding the quote from comment 2: “it is just a direct use of ADAM in the optimization process. No specific customization for BERT.”
>
> LAMB uses careful per-parameter and layerwise normalization for achieving good performance with large batches. The update rule is different from Adam: the proposed layer-wise adaptive learning rate can be found in eq(3), where g_t is the SGD or Adam update. This approach is motivated by the general layerwise adaptation strategy proposed in the paper for large batch settings. LAMB is robust and, as shown in the paper, works across a wide range of applications without much tuning or customization. This should be seen in positive light since LAMB can be used out-of-the-box for various applications without much customization or tuning. In fact, LAMB has been used for training a state-of-the-art model leading to best known results on the GLUE, RACE, and SQuAD benchmarks.
>
> We don’t intend to do any customization on BERT. Our goal is to provide a generalized optimization technique which can be applied on a wide range of models and problems.
>
> Regarding the comment (3): We would like to highlight the theoretical contributions of the paper. First, we provide a general adaptation framework specially catered for large batch settings in deep learning. LARS, which achieved SOTA results for large batch training on ResNet, is a special instantiation of the framework. Based on this framework, we propose a more robust layerwise adaptation strategy, LAMB.  Second, our paper provides convergence analysis for LARS and LAMB methods in the context of large batch learning. To our knowledge, ours is the first work to study even the convergence of LARS rigorously in the context of large batch learning for deep learning. We show that LARS and LAMB (with beta2 = 0) can converge faster than SGD (see discussion above Section 4 for more details). We believe LAMB (with beta2 > 0) should behave even better. It is an interesting and important open problem to improve the analysis given that LAMB is used for various SOTA tasks recently (again, we will be happy to provide references to these works if it does not violate double blind policy) .
>
> Overall, given the necessity of large-batch optimization for modern large-scale machine learning applications, we think this paper is very useful for both industry and academia alike.

---

### Official Review · AnonReviewer3 · 2019-10-22
**Official Blind Review #3**

**Rating:** 8

**Review:**

This paper developed a novel layerwise adaptation strategy, LAMB, that allows training BERT model with large mini-batches (32k vs baseline 512). This significantly speeds up the status quo in training BERT model, and effectively reduces the training time from original 3 days to only 76 minutes. In addition to demonstrating superior results across various tasks in practice, the paper also provides theoretical convergence analysis on LAMB optimizer.

Overall, this paper has made sound contributions that enables BERT-alike language to be trained with significant speedups, which is not otherwise achievable through LARS. The paper is well written and structured. I recommend acceptance.


**Experience Assessment:**

I have read many papers in this area.

**Review Assessment: Checking Correctness Of Derivations And Theory:**

I did not assess the derivations or theory.

**Review Assessment: Checking Correctness Of Experiments:**

I carefully checked the experiments.

**Review Assessment: Thoroughness In Paper Reading:**

I made a quick assessment of this paper.

---

> ### Author Response · Authors · 2019-11-09
> **Response to review #3**
>
> We thank the reviewer for the positive feedback and for highlighting the significance of our work.

---

### Official Review · AnonReviewer2 · 2019-10-23
**Official Blind Review #2**

**Rating:** 6

**Review:**

This paper proposes a learning rate adaptation mechanism, called LAMB, for large-batch distributed training. The goal is to stabilize the training as the batch size increases. The idea is simple and straightforward -- there should be a layerwise learning rate adjusted by normalizing the layer weights and gradients at each layer so that layers with larger weights take larger learning steps, and vice versa. The authors perform empirical studies on BERT-large and ResNet to conclude that LAMB can scale up training batch size while still being able to converge in time with comparable accuracy.

Strengths:
+ Demonstrate the scalability of large-batch training (up to 64K) on BERT-Large with comparable accuracy.
+ A leap from the prior work LARS that demonstrates the layer-wise learning rate adjustment scheme also works with Adam for NLP tasks.
+ The re-warmup technique for stabilizing the second phase of mixed sequence training is neat.

Weaknesses:
- Although the authors' analysis is based on a large set of models and clearly outperforms the prior work LARS, it is still hard to assess the generality of the obtained results. The authors made an effort to show evaluation results on MNIST and CIFAR-10, but they are much less challenging tasks.
- Technical novelty over LARS seems to be incremental, where a large portion of the work is essentially applying LARS to Adam and demonstrate its effectiveness on BERT and ResNet.

Overall, the LAMB technique seems to be simple to apply yet very useful in practice for large scale training. The work can potentially help the practitioner to scale-out large model training to hundreds or even thousands of GPUs/TPUs with good scalability. Moving forward, the authors are encouraged to report LAMB optimization results on transformer-based models such as GPT, RoBERTa, and XLNet.

Question:
Does the training take advantage of FP16 half-precision training?
How does the training process handle overflow and NaN gradients?
What is the significance of the range [α_l, α_u] in Theorem 2 and Theorem 3, and how to choose the value for them in practice?

**Experience Assessment:**

I have read many papers in this area.

**Review Assessment: Checking Correctness Of Derivations And Theory:**

I did not assess the derivations or theory.

**Review Assessment: Checking Correctness Of Experiments:**

I assessed the sensibility of the experiments.

**Review Assessment: Thoroughness In Paper Reading:**

I read the paper at least twice and used my best judgement in assessing the paper.

---

> ### Author Response · Authors · 2019-11-09
> **Response to Review #2**
>
> Thank you for giving us constructive feedback.
>
> LAMB, unlike LARS, uses both per-parameter and layerwise normalization which enables to achieve good performance for very large batches. This approach is motivated by the general layerwise adaptation strategy proposed in the paper for large batch settings. We would like to  highlight that our paper provides the first theoretical analysis for these methods in the context of large batch learning.
>
> Regarding reviewer’s comment “the authors are encouraged to report LAMB optimization results on transformer-based models such as GPT, RoBERTa, and XLNet.”
>
> LAMB has been used for training a state-of-the-art model leading to best known results on the GLUE, RACE, and SQuAD benchmarks. Another recent model has been trained using our LAMB optimizer on one GPU for 4 days that outperforms GPT (trained using 30x more compute) on the GLUE natural language understanding benchmark.  This model can also match the performance of RoBERTa while using less than 1/4 of the total compute. LAMB has also been reported to scale the transformer model to 128 GPUs without losing accuracy. It is not clear if providing the relevant works will lead to some violation of double blind policy. But we will be happy to provide references to the relevant works if this is not the case.
>
> In addition to BERT, we did not just show the results of MNIST and CIFAR, we also have ResNet-50/ImageNet results. To the best of our knowledge, all the previous adaptive optimizers fail to achieve state-of-the-art performance on ResNet-50/ImageNet task for large-batch training.
>
> Regarding the reviewer’s questions:
>
> Does the training take advantage of FP16 half-precision training?
> No, because we want to keep it the same with the BERT baseline implementation. The goal is to provide a generalized optimization technique, not to customize for a specific model.
>
> How does the training process handle overflow and NaN gradients?
> We used the gradient clipping technique which is the same as the BERT author’s implementation.
>
> What is the significance of the range [α_l, α_u] in Theorem 2 and Theorem 3, and how to choose the value for them in practice?
> These parameters ensure that the layerwise learning rates are not small or large. This is required for the convergence analysis. We chose α_l = 1e-6 and α_u was not required for all our experiments. We also observed that LAMB is quite robust to changes in these parameters.

---

### Decision · Program_Chairs · 2019-12-19

**Decision:**

Accept (Poster)

**Comment:**

This paper presents a range of methods for over-coming the challenges of large-batch training with transformer models.  While one reviewer still questions the utility of training with such large numbers of devices, there is certainly a segment of the community that focuses on large-batch training, and the ideas in this paper will hopefully find a range of uses.